# SI-IGCL: Subject Invariance-aware Inverse Graph Contrastive Learning for Psychiatric Disorder Identification

**Jiayu Lu** [1]   **Yujin Wang** [1]   **Xiaofeng Liu** [1]   **Dandan Li** [1]   **Bin Wang** [1]

## Abstract

Functional brain network analysis plays an important role in understanding and diagnosing psychiatric disorders. However, current methods struggle with subject variations, impairing the model's generalization ability to the test set. To address this issue, we propose the Subject Invariance-aware Inverse Graph Contrastive Learning (SI-IGCL) model, which adopts a two-stage paradigm with self-supervised subject-invariant pre-training followed by supervised fine-tuning for identification. During the pre-training phase, we construct an inverse contrastive objective that reshapes the embedding space by repelling intra-subject and attracting inter-subject embeddings to learn subject-invariant representations, with an auxiliary correction term to avoid early optimization plateaus. Meanwhile, we incorporate a structure-preserving reconstruction constraint to preserve discriminative information. Moreover, a Hierarchical Topology Enhanced Transformer (HTET) module is designed to enable multi-level modeling of subject-invariant functional patterns. During the fine-tuning phase, a supervised classifier is integrated to perform psychiatric disorder classification. Extensive experiments demonstrate that our method outperforms all state-of-the-art methods. The code is available at https://anonymous.4open.science/r/SI-IGCL.

## 1. Introduction

Brain psychiatric disorders such as autism spectrum disorder (ASD) and attention deficit hyperactivity disorder (ADHD) impact the quality of life for hundreds of millions of people worldwide (Lord et al., 2020; Da Silva et al., 2023). These

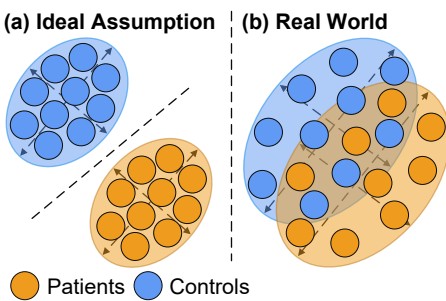

*Figure 1.* An illustration comparing the idealized group homogeneity assumption in brain functional connectivity analysis (left) with the pronounced inter-subject differences and inter-group overlap observed in real word (right).

disorders involve complex neurobehavioral and neurobiological mechanisms, making accurate diagnosis particularly challenging (Andreazza et al., 2025). Functional magnetic resonance imaging (fMRI) provides a non-invasive way to measure blood-oxygen-level-dependent (BOLD) signals and estimate functional connectivity (FC) among brain regions of interest (ROIs) (Biswal & Uddin, 2025; Li et al., 2025). Analyzing these connectivity patterns can reveal abnormal brain network organization relevant to diagnosis and treatment (Yap et al., 2024).

Despite the diagnostic potential of functional connectivity analysis (Tang et al., 2025), reliable disease-related aberrant patterns are often obscured by pronounced inter-subject variability, which acts as a dominant source of noise (Yu et al., 2025). Brain FC patterns exhibit fingerprint-like specificity (Finn et al., 2015), arising from a unique combination of genetic, developmental, and experiential factors, with variability that can even exceed the systematic alterations induced by disease (Stampacchia et al., 2024). Consequently, substantial inter-subject differences pose a major challenge for accurate psychiatric disorder diagnosis.

Several approaches based on Graph Neural Networks (GNNs) and Graph Transformers (GTs) have shown promising results for psychiatric disorder diagnosis (Peng et al., 2024a; Song et al., 2025; Pei et al., 2025). Most of these methods are designed under a group homogeneity assumption (Fig. 1), which assumes that subjects within the patient group and the control group are relatively homogeneous,

[1] College of Computer Science and Technology, Taiyuan University of Technology, Taiyuan, Shanxi, China . Correspondence to: Bin Wang <wangbin01@tyut.edu.cn>.

*Proceedings of the 43rd International Conference on Machine Learning*, Seoul, South Korea. PMLR 306, 2026. Copyright 2026 by the author(s).

and that inter-group differences are much larger than intra-group subject differences (Yu et al., 2024; Ding et al., 2025; Peng et al., 2025). However, substantial inter-subject distributional differences in brain FC data contradict this assumption. As a result, these models may rely on spurious correlations and extract fragile features from specific samples, which do not generalize well to test data.

Facing this bottleneck, we propose the Subject Invariance-aware Inverse Graph Contrastive Learning (SI-IGCL) model. SI-IGCL introduces a two-stage paradigm that separates subject-invariant pre-training from supervised fine-tuning, effectively mitigating subject variations. To encourage the encoder learning subject-invariant representations, we invert the standard contrastive objective by repelling embeddings from the same subject while attracting those across different subjects, with an auxiliary correction term to mitigate early optimization plateaus. Meanwhile, to prevent potential representation collapse, we design a structure-preserving reconstruction loss as a complementary regularization. Furthermore, we introduce the hierarchical topology enhanced transformer (HTET) module, specifically designed to capture subject-invariant patterns distributed across multiple topological levels. The principal contributions can be summarized as follows:

- We propose the first inverse graph contrastive learning framework for subject-invariant representation learning in brain networks, enabling robust psychiatric disorder identification under substantial inter-subject variability.

- We formulate an inverse contrastive objective with a structure-preserving reconstruction constraint, mitigating subject differences while preserving disease-relevant features.

- We design a hierarchical topology enhanced transformer module for capturing subject-invariant patterns across multiple topological levels.

## 2. Related Work

Existing methods for brain disorder identification based on FC mainly rely on graph neural networks (GNNs) and graph transformers (GTs). In particular, GNN-based methods have been developed with diverse methodological strategies. Some approaches aim to standardize graph construction and improve robustness. BrainGB (Cui et al., 2022) unifies neuroimaging-based pipelines and modular GNN implementations. AGE-GCN (Ding et al., 2025) enhances dissimilarities between brain regions, and GroupBNA (Peng et al., 2024a) adapts to distinct subject groups to improve robustness. Another line of work incorporates causal principles to extract disease-relevant subgraphs and improve interpretability. For instance, CIA-GCL (Yu et al., 2025)

leverages causal disentanglement and invariant learning to extract brain-invariant subgraphs, while CI-GNN (Zheng et al., 2024) adopts a Granger causality-inspired GNN for interpretable psychiatric diagnosis. Several approaches focus on capturing complex or dynamic network structures. DSVB (Yap et al., 2024) models time-varying topologies, and BrainHGL (Wen et al., 2025), STW-HCN (Liu et al., 2024b), and HSGNN (Chen et al., 2025) are designed to handle heterogeneous connectivity patterns. Finally, methods such as CRGNN (Xia et al., 2024), BrainIB (Zheng et al., 2025), LG-GNN (Zhang et al., 2023), and MAHGCN (Liu et al., 2024c) enhance diagnostic performance by integrating auxiliary information or employing adaptive pooling to comprehensively extract disease-relevant features.

Beyond GNN-based methods, GT-based approaches have also been developed for brain disorder identification (Kan et al., 2022). Several methods incorporate structural or topological priors, including METAFormer (Mahler et al., 2023), which leverages multi-atlas information, BioBGT (Peng et al., 2025), encoding small-world architectures, and CAGT (Pei et al., 2025) and Com-BrainTF (Bannadabhavi et al., 2023), which integrate community information into transformer architectures. To better capture long-range and temporally related dependencies, ALTER (Yu et al., 2024) uses biased random walks, while RGTNet (Wang et al., 2024) employs a residual graph encoder. Other methods focus on enhancing diagnostic performance by addressing data heterogeneity or selecting informative features, such as KAGT (Song et al., 2025) with domain adaptation, LC-CAF (Kang et al., 2023) with adaptive attention, Gradformer (Liu et al., 2024a) emphasizing structural inductive biases, and Contrasformer (Xu et al., 2024) using prior-knowledge-enhanced contrast graphs. GBT (Peng et al., 2024b) further combines attention approximation with geometric representation learning for comprehensive connectivity analysis.

These methods have shown promising results in improving diagnostic performance. However, they rely on a group homogeneity assumption that is often violated by substantial inter-subject differences, leading to spurious correlations and poor generalization.

## 3. Method

Fig. 2 and Fig. 3 show the pre-training and fine-tuning phases of the proposed SI-IGCL model, respectively. During the pre-training stage, subject invariance-aware representations are learned using a self-supervised inverse contrastive objective combined with a reconstruction constraint. During the fine-tuning stage, the pre-trained encoder is adapted to psychiatric disorder classification in a supervised manner.

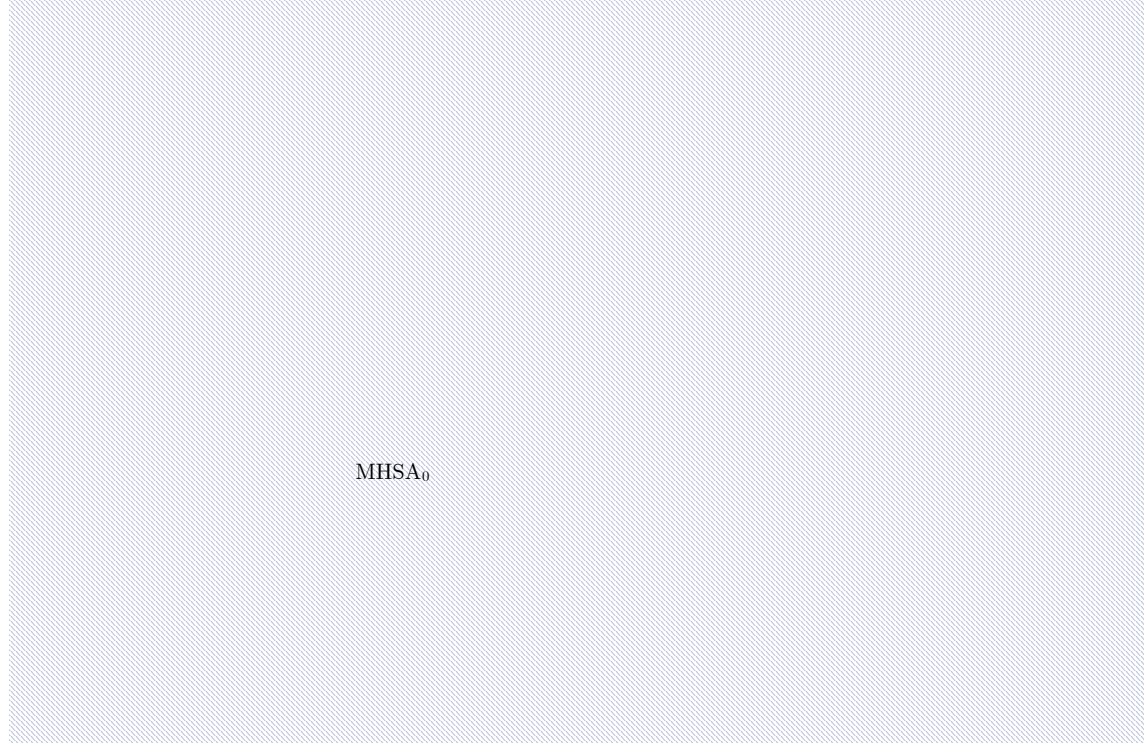

MHSA$_0$

*Figure 2.* The pre-training phase of SI-IGCL. The model extracts subject-invariant representations in a self-supervised manner using an inverse contrastive objective with a reconstruction constraint. It consists of four components including same-subject and cross-subject pair construction, the hierarchical topology enhanced transformer (HTET) module, a reconstruction module, and a subject invariance loss construction module.

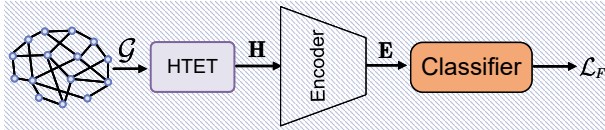

*Figure 3.* The fine-tuning phase of SI-IGCL. The pre-trained HTET and the encoder is adapted for supervised psychiatric disorder classification via readout and a linear classifier.

## 3.1. Problem Definition

We define a functional connectome as a graph $\mathcal{G} = (\mathcal{V}, \mathcal{E}, \mathbf{A})$, where the node set $\mathcal{V} = \{v_1, \cdots, v_N\}$ represents $N$ regions-of-interest (ROIs), the edge set $\mathcal{E} \subseteq \mathcal{V} \times \mathcal{V}$ encodes FC relations, and the adjacency matrix $\mathbf{A} \in \mathbb{R}^{N \times N}$ stores the corresponding connectivity strengths. Our goal is to learn a two-stage mapping consisting of a pre-training encoder $f_1$ that extracts subject-invariant embeddings $\mathbf{E}_{\mathrm{SI}}$, followed by a fine-tuning classifier $f_2$ for psychiatric disorder identification. Formally,

$$f_1 : \mathcal{G} \mapsto \mathbf{E}_{\mathrm{SI}} \in \mathbb{R}^{N \times d}, f_2 : \mathbf{E}_{\mathrm{SI}} \mapsto \hat{y} \in [0, 1], \quad (1)$$

where $d$ denotes the embedding dimension, each row of $\mathbf{E}_{\mathrm{SI}}$ corresponds to a node-level representation, and $\hat{y}$ represents the predicted probability of psychiatric disorder.

## 3.2. Pre-training toward Subject-invariant Representation

In the pre-training stage, we exploit subject identity as an intrinsic self-supervision signal to construct contrastive pairs. Each FC graph is encoded by the hierarchical topology enhanced transformer (HTET) to capture multi-level topological representations, while a reconstruction module is employed to restore the original connectivity structure from the learned embeddings. To explicitly disentangle subject variations, we formulate a subject invariance objective that integrates an inverse contrastive loss with a reconstruction constraint, where subject-invariant representations are learned by repelling intra-subject embeddings and attracting inter-subject embeddings.

### 3.2.1. SAME-SUBJECT AND CROSS-SUBJECT PAIRS CONSTRUCTION

To avoid topology distortions introduced by handcrafted graph augmentations (Ding et al., 2022), subject identity is used as the self-supervision signal to construct contrastive pairs as shown in Fig. 2a. Given a population of $M$ subjects $\mathcal{S} = \{s_1, \cdots, s_M\}$, two connectome graphs $\{\mathcal{G}_i^1, \mathcal{G}_i^2\}$ are constructed for each subject $s_i$ by splitting the fMRI time series into two non-overlapping segments (see Appendix A.1

for details). Intra-subject graphs are treated as same-subject pairs, while inter-subject graphs are treated as cross-subject pairs:

$$\mathcal{P}^+ = \left\{ (\mathcal{G}_i^1, \mathcal{G}_i^2) \mid i \in \{1, 2, \ldots, M\} \right\},$$
$$\mathcal{P}^- = \left\{ (\mathcal{G}_i^a, \mathcal{G}_j^b) \mid i \neq j, \ a, b \in \{1, 2\} \right\}. \qquad (2)$$

### 3.2.2. HIERARCHICAL TOPOLOGY ENHANCED TRANSFORMER

To fully capture subject-invariant patterns across multiple topological scales, we design a hierarchical topology enhanced transformer (HTET) module as shown in Fig. 2b. Specifically, HTET first constructs hierarchical graph structures to model multi-level topological organization. Level-specific multi-head self-attention is then employed to capture both local and global dependencies at each hierarchical level. Finally, a hierarchical fusion module aggregates representations across levels to form unified subject-invariant embeddings.

**Hierarchical Graph Construction.** To avoid the introduced evidence problem caused by learnable edge weighting and the distorted connectivity problem arising from dynamic sparsification (Yuan et al., 2022), we derive a hierarchy of subgraphs from each connectome using percentile-based adaptive thresholding (Ye et al.; Peng et al., 2025). Given $\mathcal{G} = (\mathcal{V}, \mathcal{E}, \mathbf{A})$, we construct $K$ level-specific subgraphs:

$$\mathcal{G}_k' = (\mathcal{V}, \mathcal{E}_k), \quad \mathcal{E}_k = \{(i, j) \in \mathcal{E} \mid A_{ij} \geq \theta_k\}, \quad (3)$$

where $k \in \{1, \ldots, K\}$ indexes the hierarchy, and $\theta_k$ is a level-specific threshold determined by retaining a proportion $p_k \in (0, 1]$ of the strongest edges:

$$\theta_k = \inf\left\{ t \in \mathbb{R} \ \Big| \ \sum_{1 \leq i < j \leq N} \mathbb{I}(A_{ij} \geq t) \leq p_k |\mathcal{E}| \right\}. \quad (4)$$

Here, $t$ denotes a candidate threshold on edge weights, $\mathbb{I}(\cdot)$ is the indicator function, $|\mathcal{E}|$ is the number of edges. Larger $p_k$ corresponds to denser graph views, while smaller $p_k$ focuses on the strongest connections.

**Level-specific Multi-head Self-attention Module.** For each level-specific graph $\mathcal{G}_k'$, we apply an $L$-layer multi-head self-attention (MHSA) module to obtain topology-aware node representations. Let $\mathbf{X}_k \in \mathbb{R}^{N \times N}$ denote the input node features for graph $\mathcal{G}_k'$. The MHSA output at layer $l$ is computed as:

$$\mathbf{H}_k^l = \bigg\|_{c=1}^{C} \mathbf{h}_k^{l,c} \mathbf{W}_O^{l,c},$$

$$\mathbf{h}_k^{l,c} = \text{Softmax}\left( \frac{(\mathbf{Z}_k^{l-1} \mathbf{W}_Q^{l,c})(\mathbf{Z}_k^{l-1} \mathbf{W}_K^{l,c})^\top}{\sqrt{d_K^{l,c}}} \right) \mathbf{Z}_k^{l-1} \mathbf{W}_V^{l,c}, \qquad (5)$$

where $l \in \{1, \ldots, L\}$ is the layer index, $c \in \{1, \ldots, C\}$ is the attention head index, $\mathbf{Z}_k^0 = \mathbf{X}_k$, and $\mathbf{Z}_k^{l-1}$ is the node representation matrix entering layer $l$. The matrices $\mathbf{W}_Q^{l,c}$, $\mathbf{W}_K^{l,c}$, and $\mathbf{W}_V^{l,c}$ are the learnable query, key, and value projection matrices of head $c$ at layer $l$, respectively, and $\mathbf{W}_O^{l,c}$ projects the head output to the layer output dimension $d_l$. The scalar $d_K^{l,c}$ is the key dimension used for scaling. The operator $\|$ denotes concatenation across $C$ heads. The final HTET output at level $k$ is $\mathbf{H}_k = \mathbf{H}_k^L \in \mathbb{R}^{N \times N}$.

**Hierarchical Feature Fusion.** To capture complementary functional patterns across topological scales and reinforce stable subject-invariant patterns, we fuse node embeddings from all $K$ hierarchical levels into a unified representation:

$$\mathbf{H} = \text{LayerNorm}\left( \mathbf{H}_0 + \sum_{k=1}^{K} \gamma_k \mathbf{H}_k \right), \qquad (6)$$

where $\mathbf{H}_0$ encodes the original FC graph. The residual connection with $\mathbf{H}_0$ ensures that information from the original FC graph is preserved during hierarchical processing. The coefficients $\gamma_k \in \mathbb{R}$ are learnable parameters, initialized as $\frac{1}{K}$, which adapt during training to reflect the relative importance of each level for learning subject-invariant representations. Finally, LayerNorm$(\cdot)$ normalizes the fused embeddings to stabilize training.

### 3.2.3. RECONSTRUCTION MODULE

The reconstruction module maps the fused HTET representation to subject invariance-aware embeddings and reconstructs the original adjacency to preserve topological structure. We implement the encoder as a graph neural network (GNN) that takes $\mathbf{H} \in \mathbb{R}^{N \times N}$ as input and outputs node embeddings:

$$\mathbf{E}_{\text{SI}} = \text{Encoder}(\mathbf{H}) \in \mathbb{R}^{N \times d}, \qquad (7)$$

where $d$ denotes the embedding dimension and $\mathbf{E}_{\text{SI}}$ represents the subject-invariant embeddings. A GNN-based decoder then reconstructs an adjacency-like matrix from $\mathbf{E}_{\text{SI}}$:

$$\mathbf{Z} = \text{Decoder}(\mathbf{E}_{\text{SI}}) \in \mathbb{R}^{N \times N}, \qquad (8)$$

where $\mathbf{Z}$ is the reconstructed connectivity matrix. The encoder is therefore trained to encode sufficient structural information in $\mathbf{E}_{\text{SI}}$ to enable faithful reconstruction of the input connectome, while the inverse contrastive objective introduced next acts as a regularizer that suppresses inter-subject variability.

### 3.2.4. SUBJECT INVARIANCE LOSS

we formulated a composite subject invariance loss ($\mathcal{L}_{\text{SI}}$) consisting of an inverse contrastive loss ($\mathcal{L}_{\text{IC}}$) and a recon-

struction loss ($\mathcal{L}_R$) as shown in Fig. 2c. The inverse contrastive loss repels embeddings from the same subject while attracting those across different subjects, thereby promoting subject-invariant representations. The reconstruction loss constrains the model to preserve essential graph topology, preventing degenerate solutions and representation collapse.

**Inverse Contrastive Loss.** Recall that each subject has two connectome graphs. yielding $2B$ graphs in a batch. Let $\mathcal{Z}^1 = \{\mathbf{Z}_i^1\}_{i=1}^B$ and $\mathcal{Z}^2 = \{\mathbf{Z}_i^2\}_{i=1}^B$ denote the reconstructed adjacency matrices for the two views, where $\mathbf{Z}_i^1, \mathbf{Z}_i^2 \in \mathbb{R}^{N \times N}$ are obtained from the decoder. For any pair $(i, j)$, we define their similarity using the cosine similarity induced by the Frobenius inner product (Montero et al., 2002):

$$s_{ij}^{(a,b)} = \frac{\langle \mathbf{Z}_i^a, \mathbf{Z}_j^b \rangle_F}{\|\mathbf{Z}_i^a\|_F \|\mathbf{Z}_j^b\|_F}, \quad a, b \in \{1, 2\}, \qquad (9)$$

where $\langle \mathbf{Z}_i^a, \mathbf{Z}_j^b \rangle_F = \sum_{u=1}^N \sum_{v=1}^N Z_i^a(u, v) Z_j^b(u, v)$ is the Frobenius inner product and $\|\cdot\|_F$ is the Frobenius norm.

For each anchor $\mathbf{Z}_i^1$, its same-subject counterpart is $\mathbf{Z}_i^2$, and all samples from other subjects $\{\mathbf{Z}_j^b \mid j \neq i, b \in \{1, 2\}\}$ are treated as cross-subject pairs. We design an inverse contrastive loss that reverses the standard contrastive objective (Chen et al., 2020) by encouraging higher similarity between embeddings from different subjects and lower similarity between embeddings from the same subject:

$$\mathcal{L}_{IC} = -\frac{1}{B} \sum_{i=1}^B \log \frac{\sum_{j \neq i} \sum_{a,b \in \{1,2\}} \left(E_{ij}^{(a,b)} - \alpha\right)}{E_i^+ + \sum_{j \neq i} \sum_{a,b \in \{1,2\}} \left(E_{ij}^{(a,b)} - \alpha\right)},$$

$$(10)$$

where $E_i^+ = \exp\left(s_{ii}^{(1,2)}/\tau\right)$ and $E_{ij}^{(a,b)} = \exp\left(s_{ij}^{(a,b)}/\tau\right)$. Here, $\tau > 0$ is a temperature parameter, and $\alpha > 0$ is a corrective term introduced to offset the contribution of uninformative cross-subject pairs whose similarities are already sufficiently high. By attenuating such pairs, $\alpha$ prevents them from dominating the inverse contrastive objective, thereby sustaining informative gradients and facilitating continued loss descent during training.

Optimizing $\mathcal{L}_{IC}$ alone may lead to representation collapse, where the reconstruction module maps all inputs to nearly identical reconstructed matrices. In this degenerate case, subject-level differences are eliminated at the expense of meaningful connectivity structure, yielding embeddings with limited discriminative power for downstream tasks. To avoid this issue, we introduce a reconstruction loss as a complementary constraint.

**Reconstruction Loss.** The reconstruction loss encourages the decoder output $\mathbf{Z}_i^a$ to faithfully approximate the original

adjacency matrix $\mathbf{A}_i^a$ of each input graph $\mathcal{G}_i^a$. For a batch of $B$ subjects, we define:

$$\mathcal{L}_R = \frac{1}{2B} \sum_{i=1}^B \sum_{a=1}^2 \frac{1}{N^2} \sum_{u=1}^N \sum_{v=1}^N \left(Z_i^a(u, v) - A_i^a(u, v)\right)^2,$$

$$(11)$$

where $A_i^a(u, v)$ and $Z_i^a(u, v)$ denote the $(u, v)$-th entries of the original and reconstructed adjacency matrices for subject $i$, respectively. $\mathcal{L}_R$ regularizes the inverse contrastive objective by constraining the learned embeddings to preserve essential graph topology. As a result, it prevents potential representation collapse while retaining connectivity patterns relevant for psychiatric disorder classification.

Thus, $\mathcal{L}_R$ acts as a structural prior that anchors the embeddings to the underlying connectivity structure of brain connectomes, while $\mathcal{L}_{IC}$ shapes the invariance properties with respect to subject identity. Finally, the model completes iterative training by jointly optimizing the inverse contrastive and reconstruction objectives. The overall subject invariance loss is:

$$\mathcal{L}_{SI} = \mathcal{L}_R + \beta \cdot \mathcal{L}_{IC}, \qquad (12)$$

where $\beta > 0$ is a trade-off hyperparameter that balances the strength of the inverse contrastive term against the reconstruction fidelity. By appropriately tuning $\beta$, we obtain embeddings that effectively suppress subject variability while maintaining the connectivity information required for robust psychiatric disorder prediction.

### 3.3. Fine-tuning for Downstream Classification

The fine-tuning stage aims to leverage the pre-trained subject-invariant representations to achieve more accurate psychiatric disorder classification (Fig. 3). Specifically, we reuse the pre-trained HTET and encoder to obtain node-level embeddings for each subject. Let $\mathbf{E}_i \in \mathbb{R}^{N \times d}$ denote the encoder output for subject $s_i$. We aggregate node embeddings into a graph-level representation $\mathbf{h}_i \in \mathbb{R}^d$ using a readout function:

$$\mathbf{h}_i = \text{Readout}(\mathbf{E}_i). \qquad (13)$$

A psychiatric disorder classifier, implemented as a single-layer fully connected network, is applied to $\mathbf{h}_i$ to obtain the predicted probability $\hat{y}_i \in [0, 1]$:

$$\hat{y}_i = \sigma(\mathbf{D}^\top \mathbf{h}_i), \qquad (14)$$

where $\mathbf{D}$ denotes the classifier weight vector and $\sigma(\cdot)$ is the sigmoid function. We train the classifier and optionally fine-tune all parameters of HTET and the encoder by minimizing the binary cross-entropy loss over all $M$ subjects:

$$\mathcal{L}_F = -\frac{1}{M} \sum_{i=1}^M \left[y_i \log \hat{y}_i + (1 - y_i) \log(1 - \hat{y}_i)\right], \quad (15)$$

*Table 1.* Performance (%) comparison with baselines. **Bold** indicates the best results and underlining denotes the second best outcomes.

| Method | ABIDE | | | | | ADHD-200 | | | | |
|---|---|---|---|---|---|---|---|---|---|---|
| | ACC | SEN | SPE | F1 | AUC | ACC | SEN | SPE | F1 | AUC |
| BrainGB | 71.07±4.92 | 72.90±6.20 | 68.73±7.36 | 70.80±4.47 | 74.93±5.10 | 71.91±2.89 | 45.56±9.96 | 90.47±11.95 | 54.62±6.65 | 71.63±6.30 |
| CRGNN | 52.71±10.32 | 63.32±15.50 | 42.45±18.84 | 58.71±6.42 | 52.91±5.05 | 51.65±9.78 | 62.15±16.20 | 40.78±12.98 | 56.31±5.15 | 52.78±5.56 |
| CI-GNN | 71.89±2.91 | 73.37±4.80 | 69.44±5.37 | 70.58±2.21 | 73.32±3.62 | 71.03±3.05 | 53.44±10.84 | 88.69±9.76 | 66.93±4.26 | 71.95±2.97 |
| CIA-GCL | 71.95±3.36 | 76.23±8.39 | 66.73±11.92 | 71.35±3.79 | 74.26±3.88 | 64.93±3.84 | 32.22±20.95 | 90.91±9.09 | 36.89±14.95 | 64.06±4.81 |
| BrainTrans | 71.90±2.77 | 75.17±8.45 | 68.33±9.58 | 75.20±2.84 | 78.80±2.59 | 64.08±4.12 | 28.42±9.18 | 86.24±7.09 | 36.91±8.77 | 66.80±4.69 |
| Com-BrainTF | 70.14±4.38 | 72.83±4.15 | 67.38±4.75 | 70.01±4.49 | 71.67±6.16 | 71.78±4.50 | 56.71±17.17 | 85.76±16.81 | 68.28±7.69 | 69.75±6.96 |
| METAFormer | 70.31±2.86 | 74.38±6.64 | 67.58±6.48 | 72.85±3.29 | 72.29±3.54 | 70.27±2.54 | 44.09±10.01 | 91.38±12.48 | 54.66±9.45 | 69.13±4.65 |
| Contrasformer | 68.90±2.33 | 70.91±6.04 | 65.47±7.94 | 68.70±2.74 | 70.68±2.64 | 72.19±2.65 | 59.53±12.73 | 87.91±13.49 | 60.26±7.32 | 72.63±3.41 |
| RGTNet | 69.52±3.51 | 70.55±4.54 | 70.00±4.52 | 71.51±2.65 | 71.05±2.45 | 60.23±2.84 | 68.00±2.50 | 43.40±4.55 | 47.80±3.45 | 58.30±2.20 |
| BrainIB | 69.97±2.82 | 70.70±4.61 | 69.77±5.27 | 70.74±2.90 | 73.44±4.35 | 70.07±1.56 | 56.47±8.48 | 85.32±8.14 | 58.10±4.11 | 67.28±2.72 |
| GBT | 70.06±4.96 | 73.08±7.73 | 66.24±10.05 | 72.86±2.15 | 75.80±3.79 | 66.84±2.81 | 44.50±17.58 | 80.86±9.99 | 61.77±5.28 | 72.15±4.68 |
| ALTER | 70.80±4.12 | 72.68±10.24 | 68.49±9.96 | 73.61±5.52 | 78.70±2.70 | 62.76±4.04 | 33.38±17.46 | 81.96±9.67 | 37.84±14.24 | 68.51±2.33 |
| LCCAF | 71.72±1.45 | 76.71±7.45 | 65.16±7.91 | 71.45±1.55 | 68.91±2.98 | 57.45±4.21 | 56.12±7.42 | 65.13±7.94 | 60.40±3.25 | 55.90±3.86 |
| CAGT | 71.28±1.83 | 70.37±10.37 | 68.00±10.43 | 72.38±3.24 | 71.13±5.42 | 71.65±2.34 | 73.14±4.81 | 69.83±6.63 | 71.46±2.45 | 75.46±2.94 |
| **SI-IGCL** | **75.27±2.30** | **82.25±5.47** | **72.76±4.93** | **77.27±2.65** | **79.55±4.99** | **74.26±2.68** | **73.93±3.07** | **95.55±4.84** | **73.36±2.65** | **78.86±1.46** |

*Table 2.* Comparison of cross-site generalization performance (%) with baselines. **Bold** indicates the best results.

| Method | ABIDE | | | | | ADHD-200 | | | | |
|---|---|---|---|---|---|---|---|---|---|---|
| | ACC | SEN | SPE | F1 | AUC | ACC | SEN | SPE | F1 | AUC |
| BrainGB | 67.83±4.52 | 68.75±6.83 | 66.28±7.58 | 67.29±5.14 | 72.46±5.27 | 67.24±4.82 | 66.53±7.26 | 65.84±8.16 | 66.78±5.54 | 70.83±5.68 |
| CRGNN | 49.82±5.67 | 57.46±16.28 | 38.72±19.53 | 55.39±7.63 | 50.18±5.82 | 50.13±5.24 | 56.31±15.76 | 40.21±18.94 | 54.85±7.23 | 51.24±5.57 |
| CI-GNN | 68.96±3.42 | 70.28±5.63 | 67.05±6.14 | 68.75±3.07 | 71.45±4.27 | 69.24±3.12 | 70.16±5.43 | 67.31±5.93 | 68.95±2.81 | 72.15±4.06 |
| CIA-GCL | 69.12±3.81 | 72.68±9.24 | 64.31±12.57 | 68.72±4.35 | 72.43±4.51 | 69.56±3.54 | 72.32±8.91 | 65.23±11.84 | 69.03±4.13 | 73.16±4.35 |
| BrainTrans | 69.75±1.78 | 65.60±7.75 | 69.83±6.66 | 70.24±3.65 | 77.82±0.99 | 61.28±2.54 | 59.87±7.53 | 63.52±6.48 | 60.89±3.44 | 70.56±1.52 |
| Com-BrainTF | 67.24±4.21 | 68.91±4.93 | 65.17±5.28 | 67.52±5.03 | 69.73±6.38 | 67.24±5.41 | 65.17±6.78 | 67.11±4.22 | 69.45±4.93 |
| METAFormer | 67.48±3.42 | 69.56±7.31 | 65.32±7.15 | 69.27±4.18 | 70.14±4.27 | 68.15±3.12 | 68.96±6.74 | 66.53±7.38 | 68.58±3.82 | 70.52±4.16 |
| Contrasformer | 66.37±2.87 | 67.85±6.93 | 63.92±8.57 | 66.58±3.42 | 68.74±3.21 | 67.52±2.61 | 66.79±6.54 | 65.34±7.91 | 66.92±3.12 | 69.14±3.53 |
| RGTNet | 66.93±3.87 | 67.28±5.12 | 67.42±5.37 | 68.73±3.28 | 68.95±3.06 | 66.41±3.52 | 66.21±5.03 | 66.84±5.83 | 67.83±3.02 | 69.32±3.27 |
| BrainIB | 67.63±3.45 | 67.59±5.38 | 67.14±6.02 | 68.19±3.76 | 71.38±4.93 | 67.95±3.21 | 67.13±5.14 | 67.56±6.31 | 68.43±3.52 | 71.82±4.64 |
| GBT | 67.38±5.14 | 69.73±8.42 | 63.57±10.82 | 69.27±2.93 | 73.67±4.16 | 61.78±4.83 | 60.31±8.24 | 60.25±10.53 | 61.12±2.81 | 70.23±4.03 |
| ALTER | 67.96±4.57 | 69.15±10.83 | 66.28±10.47 | 70.84±6.31 | 76.83±3.15 | 62.14±4.23 | 61.54±10.54 | 61.87±10.23 | 61.93±6.04 | 70.86±3.07 |
| LCCAF | 68.73±2.06 | 72.15±8.27 | 62.39±9.14 | 68.43±2.48 | 66.82±3.37 | 68.21±2.32 | 71.54±8.03 | 63.23±8.71 | 68.02±2.72 | 67.24±3.85 |
| CAGT | 68.39±2.64 | 66.95±10.82 | 65.28±10.94 | 69.35±4.16 | 69.27±6.08 | 68.81±2.41 | 67.24±10.54 | 66.13±10.83 | 69.14±4.03 | 69.83±5.74 |
| **SI-IGCL** | **71.45±6.71** | **74.27±16.67** | 68.45±20.50 | **71.78±7.57** | **78.17±9.34** | **70.36±6.23** | 70.58±16.52 | 66.87±20.04 | **69.25±1.34** | **75.68±9.08** |

where $y_i \in \{0, 1\}$ is the ground-truth label for subject $s_i$. Through supervised fine-tuning, the subject-invariant representations learned during pre-training are adapted to form discriminative decision boundaries between patients and healthy controls.

# 4. Experiments

## 4.1. Comparison with SOTA Methods

**Overall Performance.** Due to page limitations, we provide detailed experimental settings in Appendix A.1, including dataset descriptions and preprocessing procedures, evaluation metrics, baseline methods, and implementation details. The experimental results are summarized in Tab. 1 on the two datasets. SI-IGCL significantly outperforms all SOTA methods across all evaluation metrics. On ABIDE, it surpasses the second-best method (CIA-GCL) by 3.32%, and on ADHD-200, it exceeds the second-best method (Contrasformer) by 2.07%. These results highlight the effectiveness of our two-stage subject-invariant learning paradigm with inverse contrastive pre-training for robust psychiatric diagnosis. We further conduct multi-class diagnostic ex-

periments on ADHD-200 and ADNI to evaluate the generalization of SI-IGCL in more complex diagnostic settings. Detailed results are provided in Appendix A.2.

**Cross-site Generalization.** To further evaluate the generalization ability of SI-IGCL, we conducted leave-one-site-out (LOSO) cross-validation on both datasets. The comparison results are reported in Tab. 2. Under this challenging experimental setting, the performance of most baseline methods degrades noticeably, reflecting the substantial heterogeneity across multi-site brain data. Nevertheless, SI-IGCL still performed the best on most metrics. These results further confirm that explicitly decoupling subject-invariant pre-training from downstream adaptation substantially improves model generalization.

**Computational Efficiency Analysis.** To examine whether the proposed two-stage paradigm introduces additional computational overhead, we compare the convergence time of SI-IGCL with baselines. Specifically, we measure the total training time required to reach the target performance, computed as the number of training epochs multiplied by the per-epoch running time. For SI-IGCL, the reported training time

*Table 3.* Ablation study of SI-IGCL.

| Module | ABIDE | | | | | ADHD-200 | | | | |
|---|---|---|---|---|---|---|---|---|---|---|
| | ACC | SEN | SPE | F1 | AUC | ACC | SEN | SPE | F1 | AUC |
| w/o HTET | 71.30±5.16 | 75.88±6.46 | 66.29±7.99 | 73.00±6.02 | 69.68±5.62 | 72.32±5.92 | 70.36±5.84 | 92.03±4.84 | 70.36±4.43 | 74.80±4.73 |
| w/o $\mathcal{L}_R$ | 51.15±1.98 | 80.00±40.00 | 20.00±40.00 | 54.63±27.33 | 50.34±9.62 | 66.02±2.89 | 2.86±5.71 | **99.29±2.14** | 5.00±10.00 | 59.08±12.14 |
| w/o $\mathcal{L}_{IC}$ | 72.35±5.25 | 77.25±7.49 | 69.10±5.41 | 74.33±7.03 | 73.59±7.85 | 72.21±5.67 | 71.50±5.33 | 93.26±4.92 | 71.96±3.86 | 75.41±3.65 |
| **SI-IGCL** | **75.27±2.30** | **82.25±5.47** | **72.76±4.93** | **77.27±2.65** | **79.55±4.99** | **74.26±2.68** | **73.93±3.07** | 95.55±4.84 | **73.36±2.65** | **78.86±1.46** |

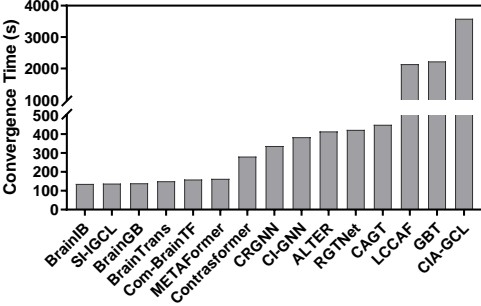

*Figure 4.* Convergence time comparison with baselines on ABIDE. Methods are ranked in ascending order.

includes both the pre-training and fine-tuning stages. The experimental results are presented in Fig. 4, where methods are ranked in ascending order of convergence time. SI-IGCL achieves the second-best training efficiency, demonstrating that the two-stage paradigm is computationally efficient and enables faster convergence through explicit subject-invariant pre-training.

### 4.2. Ablation Study and Hyperparameter Analysis

**Ablation Study on SI-IGCL.** We conducted a series of ablation studies to validate the effectiveness of each component in SI-IGCL. Specifically, we replace the proposed HTET module with a single-level Transformer to evaluate the impact of hierarchical topology modeling. Moreover, we ablate the subject invariance loss $\mathcal{L}_{SI}$ by removing the inverse contrastive loss $\mathcal{L}_{IC}$ and the reconstruction loss $\mathcal{L}_R$, respectively, to assess their individual contributions. As shown in Tab. 3, each modification results in performance degradation, highlighting the importance of all components. In particular, removing the reconstruction loss leads to a severe performance drop, and optimizing $\mathcal{L}_{IC}$ alone causes representation collapse, indicating the necessity of the reconstruction constraint for stable and discriminative representation learning.

**Effectiveness of Subject-Invariant Modeling.** To evaluate the effectiveness of subject-invariant modeling, we quantify subject variability using identifiability metrics (Lu et al., 2024). Specifically, we compute an identifiability matrix based on inter-subject FC correlations, where diagonal and off-diagonal entries denote matched and unmatched subject

*Table 4.* Effectiveness of subject-invariant modeling measured by identifiability metrics.

| Type | ABIDE | | | | ADHD-200 | | | |
|---|---|---|---|---|---|---|---|---|
| | $I_s$ | $I_o$ | $I_d$ | SIA | $I_s$ | $I_o$ | $I_d$ | SIA |
| Original | 0.7881 | 0.5712 | 0.2169 | 98.18 | 0.5672 | 0.3248 | 0.2424 | 94.16 |
| Encoder | 0.7359 | 0.6217 | 0.1142 | 79.21 | 0.4712 | 0.3707 | 0.1005 | 78.59 |
| Decoder | 0.7987 | 0.5891 | 0.2096 | 96.04 | 0.5804 | 0.3688 | 0.2116 | 85.92 |

*Table 5.* Discriminative power validation. ↓ indicates lower is better, ↑ indicates higher is better. **Bold** indicates the best results.

| Module | ABIDE | | | ADHD-200 | | |
|---|---|---|---|---|---|---|
| | NND (↓) | AFV (↑) | CKA (↑) | NND (↓) | AFV (↑) | CKA (↑) |
| Raw FC | 0.4733 | 0.2604 | 0.0502 | 0.4476 | 0.1471 | 0.0741 |
| w/o $\mathcal{L}_R$ | 0.4935 | 0.0001 | 0.0310 | 0.4800 | 0.0001 | 0.0553 |
| w/o $\mathcal{L}_{IC}$ | 0.4100 | 0.4207 | 0.2261 | 0.4381 | 0.2112 | 0.1654 |
| **SI-IGCL** | **0.3800** | **0.4309** | **0.2423** | **0.3905** | **0.3004** | **0.1775** |

similarities, respectively. We define $I_s$ and $I_o$ as the average diagonal and off-diagonal correlations, and $I_d = I_s - I_o$ to measure subject identifiability. Subject identification accuracy (SIA) is computed by assigning each subject to the counterpart with the highest correlation, and the overall percentage (%) of correct matches across all subjects is reported as the final accuracy. As shown in Tab. 4, the original FC graphs exhibit high SIA, indicating substantial inter-subject variability. After encoding, $I_d$ and SIA decrease markedly, demonstrating effective suppression of inter-subject variability. Following decoding, SIA partially increases, indicating improved reconstruction fidelity. Overall, these results confirm that our model effectively mitigates inter-subject variability.

**Discriminative Power of Pre-trained Representations.** To empirically evaluate whether the pre-trained encoder preserves disease-discriminative information, we assess the discriminative power of learned representations using nearest-neighbor diversity (NND), average feature variance (AFV), and centered kernel alignment (CKA). Detailed computation procedures are provided in Appendix A.3. As reported in Tab. 5, the proposed composite subject invariance loss $\mathcal{L}_{SI}$, consisting of the inverse contrastive loss $\mathcal{L}_{IC}$ and the reconstruction loss $\mathcal{L}_R$, substantially improves discriminative power compared with the original FC feature space. Removing $\mathcal{L}_R$ leads to representation collapse and severely degrades discriminative performance. By incorporating the

*Table 6.* Disease-related information retention on ABIDE and ADHD-200.

| Dataset | LCR ($\uparrow$) | MIR ($\uparrow$) |
|---------|--------|--------|
| ABIDE | 1.428 | 1.529 |
| ADHD-200 | 1.663 | 1.227 |

reconstruction constraint, SI-IGCL effectively avoids degenerate solutions and preserves disease-discriminative information while enforcing subject invariance. We further introduce two additional metrics: linear classifier retention (LCR) and mutual information retention (MIR) (Pohl et al., 2026). LCR measures the ratio of accuracy gains over chance achieved by the same linear classifier on the learned representation relative to raw FC, reflecting the preservation of linear separability. MIR is defined as the ratio between the mutual information of the learned representation and disease labels and that of raw FC and labels, estimated using a k-nearest neighbor estimator. Values greater than 1 indicate that disease-relevant information is preserved or enhanced. As summarized in Tab. 6, both LCR and MIR consistently exceed 1 on ABIDE and ADHD-200, indicating that disease-related information is further strengthened in the learned representation. Additional theoretical analysis of disease information preservation is provided in Appendix A.4.

**Hyperparameter Analysis.** We conduct a comprehensive hyperparameter analysis of SI-IGCL, with detailed results provided in the appendix due to space limitations. Specifically, in the HTET module, we investigate the impact of hierarchical depth and edge retention ratio (Appendix A.8). We further evaluate the runtime and memory overhead introduced by HTET to assess its computational efficiency (Appendix A.9). For the composite subject invariance loss $\mathcal{L}_{\text{SI}}$, we analyze the effects of the balancing coefficient $\beta$, the temperature parameter $\tau$, and the corrective term $\alpha$ (Appendix A.10). To verify the effectiveness of the corrective term in mitigating early optimization plateaus, we visualize the training loss trajectories with and without $\alpha$ as shown in Fig. 5. Without $\alpha$, the training process quickly falls into an early convergence plateau, indicating suboptimal optimization dynamics. In contrast, incorporating $\alpha$ provides more stable and informative gradient signals, enabling smoother optimization and improved convergence behavior. In addition, we examine the impact of the multi-head self-attention (MHSA) depth and the number of attention heads (Appendix A.11), as well as the depth and hidden dimension configurations of the reconstruction module (Appendix A.12).

### 4.3. Interpretability Analysis

We conducted an interpretability analysis to assess the biomarker detection capability of SI-IGCL. Following

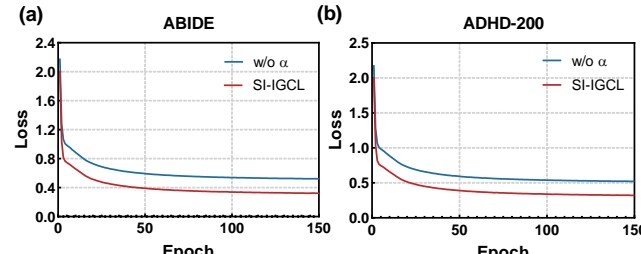

*Figure 5.* Effect of the corrective term $\alpha$ on training dynamics.

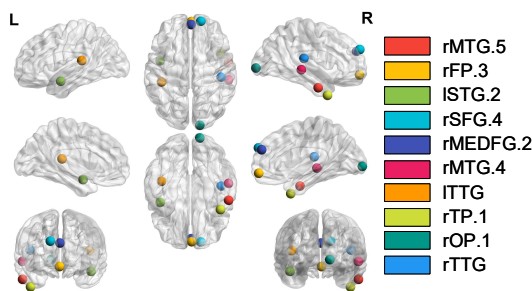

*Figure 6.* Top 10 important ROIs of ASD.

Lawrence et al. (2021); Pei et al. (2025), brain regions were grouped into eight functional subnetworks, including Cerebellum and Subcortical Structures (CB & SC), Visual Network (VN), Somatomotor Network (SMN), Dorsal Attention Network (DAN), Ventral Attention Network (VAN), Limbic Network (LN), Frontoparietal Network (FPN), and Default Mode Network (DMN). Taking ABIDE as an example, we extracted attention scores to identify salient brain regions. Fig. 6 and Tab. 7 present the top ten brain regions with the highest learned attention scores. Specifically, the DMN exhibits the highest attention weights, aligning with prior neuroscience studies, which have reported altered functional organization of the DMN and its atypical developmental trajectory as prominent neurobiological features of ASD (Padmanabhan et al., 2017; Fu et al., 2022). Moreover, multiple highly ranked regions are located in the SMN, which plays a critical role in sensory and motor processing. Previous studies have demonstrated SMN abnormalities in ASD, reflecting disrupted sensorimotor integration and atypical perceptual–motor coordination (Jung et al., 2019). Collectively, these findings demonstrate that SI-IGCL captures biologically meaningful subject-invariant functional patterns.

## 5. Conclusion and Future Work

In this work, we proposed SI-IGCL, a two-stage subject invariance-aware inverse graph contrastive learning framework for robust psychiatric disorder diagnosis. By decoupling subject-invariant pre-training from downstream adaptation and introducing an inverse contrastive objective with

*Table 7.* Details on top 10 important ROIs of ASD. "No." represents the descending sorting order, "Label" is the default order of ROI in the atlas.

| No. | Label | Network | ROI |
|---|---|---|---|
| 1 | 139 | Default Mode Network | rMTG.5 |
| 2 | 108 | Default Mode Network | rFP.3 |
| 3 | 128 | Ventral Attention Network | lSTG.2 |
| 4 | 192 | Default Mode Network | rSFG.4 |
| 5 | 90 | Default Mode Network | rMEDFG.2 |
| 6 | 106 | Somatomotor Network | rMTG.4 |
| 7 | 199 | Somatomotor Network | lTTG |
| 8 | 31 | Limbic Network | rTP.1 |
| 9 | 43 | Visual Network | rOP.1 |
| 10 | 184 | Somatomotor Network | rTTG |

reconstruction regularization, SI-IGCL effectively mitigates subject variability while preserving discriminative representations. The hierarchical topology enhanced transformer module further captures invariant functional patterns across multiple topological levels. Extensive experiments demonstrate superior performance, strong cross-site generalization, computational efficiency, and biologically meaningful interpretability, highlighting the potential of SI-IGCL for reliable neuroimaging-based diagnosis. In future work, we will extend SI-IGCL to multi-modal neuroimaging data and broader brain network analysis tasks to enhance its clinical applicability.

## Acknowledgements

Thanks for the professional and constructive suggestions from the three reviewers. This work was supported in part by the National Natural Science Foundation of China under Grants 62176177 and 62576240, in part by the Science and Technology Cooperation and Exchange Special Projects of Shanxi under grant 202304041101034, in part by the Shanxi Province Higher Education Scientific and Technological Innovation Project under grant Z2025521, in part by the Shanxi Province Application Basic Research Plan under Grant 202303021211055.

## Impact Statement

As the research in this paper focuses on the diagnosis of Autism Spectrum Disorder (ASD) and Attention Deficit Hyperactivity Disorder (ADHD), it is important to consider the potential negative social impacts of this work, even though the current research remains at a scientific stage and has not been applied in practice. One major concern is incorrect diagnosis. AI-based methods are inherently prone to errors, which cannot be entirely eliminated. An erroneous diagnosis could have serious consequences for individuals and society. Therefore, AI tools should be used solely as diagnostic aids rather than decision-makers, with final clinical judgments always made by qualified medical professionals.

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

# A. Appendix

## A.1. Experimental Settings

**Datasets and Preprocessing.** We conduct experiments on two widely used benchmark datasets: Autism Brain Imaging Data Exchange (ABIDE)[1] and Attention Deficit Hyperactivity Disorder (ADHD-200)[2]. ABIDE includes 1,009 subjects (516 ASD patients and 493 controls, ages 5–64), while ADHD-200 comprises 685 subjects (243 ADHD patients and 442 controls, ages 7–21). In both datasets, regions of interest (ROIs) are defined using the Craddock 200 atlas (Craddock et al., 2012), resulting in 200 ROIs for ABIDE and 190 ROIs for ADHD-200. To construct same-subject samples for contrastive learning, inspired by Peng et al. (2022), we split the fMRI time series of each ROI for a given subject into two non-overlapping halves. Each half is independently used to construct an FC graph by computing pearson correlations between ROI time series.

**Metrics.** In this study, all methods are evaluated using a 10-fold cross-validation protocol with consistent training–testing splits. Performance is measured by five metrics: classification accuracy (ACC), sensitivity (SEN), specificity (SPE), F1 score (F1), and ROC-AUC (AUC), where higher values reflect better outcomes. Results are reported as the mean and standard deviation across 10 independent runs of 10-fold cross-validation.

**Baselines.** We compare our model with state-of-the-art (SOTA) methods: (1) graph neural network models, including BrainGB (Cui et al., 2022), CRGNN (Xia et al., 2024), CI-GNN (Zheng et al., 2024) and CIA-GCL (Yu et al., 2025); (2) graph transformer models, including BrainTrans (Kan et al., 2022), Com-BrainTF (Bannadabhavi et al., 2023), METAFormer (Mahler et al., 2023), Contrasformer (Xu et al., 2024), RGTNet (Wang et al., 2024), BrainIB (Zheng et al., 2025), GBT (Peng et al., 2024b), ALTER (Yu et al., 2024), LCCAF (Kang et al., 2023) and CAGT (Pei et al., 2025). All publicly available methods above are compared using their original code implementations.

**Implementation Details.** The SI-IGCL model is optimized using the Adam optimizer with a StepLR learning rate scheduler. During the pre-training stage, the learning rate is set to $5 \times 10^{-4}$ with a batch size of 128. Training is conducted for up to 300 epochs with an early-stopping strategy (Prechelt, 2002). In the HTET module, the edge retention ratio is set to $p = \{0.05, 0.15\}$. For the self-attention module, the number of nonlinear mapping layers and attention heads are set to 1 and 4 for ABIDE, and 1 and 2 for ADHD-200, respectively. The encoder–decoder architecture is configured as 200–100–200 for ABIDE and 190–100–190 for ADHD-200. The overall loss function includes a balance coefficient $\beta = 0.85$, a temperature parameter $\tau = 0.1$, and a corrective term coefficient $\alpha = 1$. During fine-tuning, the learning rate is reduced to $1 \times 10^{-4}$ with a batch size of 32, and training is performed for up to 300 epochs with early stopping (Prechelt, 2002). All hyperparameters are selected through systematic tuning to ensure optimal performance. All experiments are implemented in PyTorch and conducted on a single NVIDIA RTX 4090 GPU with 48 GB memory.

## A.2. Multi-Class Generalization Evaluation

To further evaluate the generalization ability of SI-IGCL beyond binary diagnosis, we conduct additional multi-class diagnostic experiments on ADHD-200 and ADNI. Specifically, we perform four-class classification on ADHD-200, including typically developing controls, ADHD Combined type, ADHD Inattentive type, and ADHD Hyperactive/Impulsive type. We further conduct three-class classification on the Alzheimer's Disease Neuroimaging Initiative (ADNI; https://adni.loni.usc.edu/) dataset, which includes 442 subjects, consisting of 175 normal controls, 153 mild cognitive impairment patients, and 114 Alzheimer's disease patients. All methods are evaluated using a 10-fold cross-validation protocol, and performance is measured in terms of classification accuracy and Macro-F1 score. As summarized in Tab. 8, SI-IGCL achieves the best performance on most metrics across both datasets. These results demonstrate that the proposed framework generalizes well across datasets and to more complex multi-class scenarios.

## A.3. Computation of Discriminative Power Metrics

**Nearest-Neighbor Diversity (NND).** Nearest-neighbor diversity (NND) measures local discriminability by evaluating whether nearest neighbors in the feature space share the same class label (Jain et al., 2004). Given a dataset with $N$ samples

---

[1]https://fcon_1000.projects.nitrc.org/indi/abide/
[2]https://fcon_1000.projects.nitrc.org/indi/adhd200/

*Table 8.* Comparison of multi-class diagnostic performance (%) on ADHD-200 and ADNI. **Bold** indicates the best results.

| Method | ADHD-200 | | ADNI | |
|---|---|---|---|---|
| | ACC | Macro-F1 | ACC | Macro-F1 |
| BrainGB | 65.17±2.83 | 62.24±3.91 | 60.35±1.97 | 58.78±2.34 |
| CRGNN | 46.58±3.01 | 48.33±3.86 | 44.67±1.95 | 46.11±2.48 |
| CI-GNN | 64.32±2.76 | 57.85±3.71 | 60.71±1.92 | 60.44±2.26 |
| CIA-GCL | 58.73±2.88 | 52.51±3.78 | 60.28±2.01 | 58.52±2.44 |
| BrainTrans | 58.04±2.94 | 52.27±3.84 | 60.58±1.96 | 59.46±2.38 |
| Com-BrainTF | 64.95±2.80 | 59.73±3.74 | 59.36±1.89 | 56.51±2.31 |
| METAFormer | 63.62±2.72 | 50.08±3.66 | 59.63±1.93 | 58.77±2.29 |
| Contrasformer | 65.43±2.79 | 53.29±3.70 | 58.34±1.91 | 55.41±2.37 |
| RGTNet | 54.60±2.97 | 54.41±3.89 | 58.79±2.00 | 57.89±2.42 |
| BrainIB | 63.41±2.77 | 51.62±3.68 | 59.22±1.88 | 57.17±2.33 |
| GBT | 60.62±2.84 | 54.47±3.75 | 59.34±2.04 | 58.58±2.39 |
| ALTER | 56.78±2.91 | 54.89±3.82 | 60.07±1.95 | 59.42±2.41 |
| LCCAF | 51.96±2.86 | 53.80±3.76 | 60.42±1.92 | 57.92±2.36 |
| CAGT | 64.98±2.74 | **64.71±3.67** | 60.31±1.94 | 58.63±2.32 |
| **SI-IGCL** | **67.43±2.71** | 64.32±3.85 | **63.58±2.03** | **62.15±2.46** |

and extracted feature representations $\{\mathbf{f}_i\}_{i=1}^N$, we compute pairwise cosine distances

$$d_{ij} = 1 - \frac{\mathbf{f}_i \cdot \mathbf{f}_j}{\|\mathbf{f}_i\|\|\mathbf{f}_j\|}, \quad i \neq j. \tag{16}$$

For each sample $i$, we identify its $k$ nearest neighbors $\mathcal{N}_i$ (with $k = 5$) and compute the proportion of neighbors belonging to a different class:

$$\text{NND} = \frac{1}{N} \sum_{i=1}^N \frac{1}{k} \sum_{j \in \mathcal{N}_i} \mathbb{I}(y_j \neq y_i), \tag{17}$$

where $\mathbb{I}(\cdot)$ is the indicator function. Lower NND values indicate higher local discriminability, while random guessing in a balanced binary task yields NND $\approx 0.5$.

**Average Feature Variance (AFV).** Average feature variance (AFV) quantifies the utilization of the feature space and detects dimensional collapse (Bocher & McCloy, 2006). Given a feature matrix $\mathbf{F} \in \mathbb{R}^{N \times D}$, we compute the variance of each feature dimension as

$$\sigma_d^2 = \frac{1}{N} \sum_{i=1}^N (F_{id} - \mu_d)^2, \quad \mu_d = \frac{1}{N} \sum_{i=1}^N F_{id}, \tag{18}$$

and define AFV as

$$\text{AFV} = \frac{1}{D} \sum_{d=1}^D \sigma_d^2. \tag{19}$$

Higher AFV indicates better utilization of the representation space, whereas extremely low AFV suggests representation collapse.

**Centered Kernel Alignment (CKA).** Centered kernel alignment (CKA) measures the alignment between feature similarity and label similarity (Cortes et al., 2012). Given the feature kernel matrix $\mathbf{K} = \mathbf{F}\mathbf{F}^T$ and the label kernel matrix $\mathbf{L}$ defined by

$$\mathbf{L}_{ij} = \mathbb{I}(y_i = y_j), \tag{20}$$

we apply double-centering using $\mathbf{H} = \mathbf{I}_N - \frac{1}{N}\mathbf{1}_N\mathbf{1}_N^T$ to obtain $\mathbf{K}_c = \mathbf{H}\mathbf{K}\mathbf{H}$ and $\mathbf{L}_c = \mathbf{H}\mathbf{L}\mathbf{H}$. The CKA score is then computed as

$$\text{CKA}(\mathbf{K}, \mathbf{L}) = \frac{\langle \mathbf{K}_c, \mathbf{L}_c \rangle_F}{\|\mathbf{K}_c\|_F \|\mathbf{L}_c\|_F}, \tag{21}$$

where $\langle \cdot, \cdot \rangle_F$ and $\| \cdot \|_F$ denote the Frobenius inner product and norm, respectively. CKA values range from 0 to 1, with higher values indicating stronger alignment between learned representations and class labels.

### A.4. Theoretical Analysis of Disease Information Preservation

We provide a derivation based on similarity evolution and gradient competition in the loss to show that disease-relevant patterns are preserved without relying on the orthogonality assumption. We consider three types of pairs: same-subject pairs $P_{\text{same}}$ with similarity $S_{\text{same}}$, cross-subject intra-class pairs $P_{\text{within}}$ with similarity $S_{\text{within}}$, and cross-subject inter-class pairs $P_{\text{between}}$ with similarity $S_{\text{between}}$. The SI-IGCL objective consists of two components: an inverse contrastive loss $\mathcal{L}_{\text{IC}}$ that repels same-subject pairs while attracting all cross-subject pairs, and a reconstruction loss $\mathcal{L}_{\text{R}}$ that recovers the input graph to prevent representation collapse. Prior studies and empirical observations consistently show that the average similarity of same-subject pairs exceeds that of cross-subject pairs (Finn et al., 2015), with

$$\mathbb{E}[S_{\text{same}}^{(0)}] > \mathbb{E}[S_{\text{cross}}^{(0)}], \tag{22}$$

and that the average similarity of cross-subject intra-class pairs is higher than that of cross-subject inter-class pairs (Fu et al., 2023), with

$$\mathbb{E}[S_{\text{within}}^{(0)}] > \mathbb{E}[S_{\text{between}}^{(0)}], \tag{23}$$

in the raw FC space.

For each anchor $z_i$, the gradient of $\mathcal{L}_{\text{IC}}$ with respect to $z_i$ consists of two components. The repulsive term from same-subject pairs pushes $z_i$ away from its paired view and typically dominates the update due to their high initial similarity, thereby reducing reliance on subject-specific features. The attractive term pulls $z_i$ toward representations $z_j$ from other subjects, with weights determined by the current similarity. After suppressing subject-specific components, cross-subject similarity is primarily governed by disease-related features.

Let $\mu_w(t)$ and $\mu_b(t)$ denote the mean similarity of cross-subject intra-class and inter-class pairs at time $t$, with $\mu_w(0) > \mu_b(0)$. This reflects that intra-class pairs have higher initial mean similarity and therefore receive larger attraction weights. The gradient dynamics are given by

$$\frac{d\mu_w}{dt} \approx \eta(1 - \mu_w)w_w, \tag{24}$$

and

$$\frac{d\mu_b}{dt} \approx \eta(1 - \mu_b)w_b, \tag{25}$$

where $\eta$ denotes the learning rate, and $w_w$ and $w_b$ represent the effective gradient weights of cross-subject intra-class and inter-class pairs in the inverse contrastive loss. The weight ratio satisfies

$$\frac{w_w}{w_b} = e^{(\mu_w - \mu_b)/\tau}. \tag{26}$$

Therefore, the weight ratio grows exponentially with the similarity gap $\mu_w - \mu_b$, implying that the growth dynamics are dominated by the weighting mechanism. Since $\mu_w(0) > \mu_b(0)$, we have $w_w > w_b$ at initialization. As training proceeds, the increasing gap $\mu_w - \mu_b$ further amplifies the weight ratio, causing $\mu_w$ to grow consistently faster than $\mu_b$.

The reconstruction loss enforces the recovery of the original graph structure and serves as a regularizer that promotes intra-class compactness while preserving sufficient within-class variability, thereby preventing representational collapse without undermining inter-class separability. The theoretical analysis above indicates that our objective further improves the separability of disease patterns.

### A.5. Validation of The Meaningfulness and Clinical Robustness of Learned Representations

To further validate the meaningfulness and clinical robustness of the learned subject-invariant representations, we conduct scale prediction experiments on ABIDE and ADHD-200. For ABIDE, we evaluate on the Autism Diagnostic Interview-Revised (ADI-R) including Social Interaction (ADI-R Social), Verbal Communication (ADI-R Verbal), and Restricted

*Table 9.* Scale prediction performance (correlation coefficient). **Bold** indicates the best results.

| Module | ABIDE | | | | | | | ADHD-200 | | |
|---|---|---|---|---|---|---|---|---|---|---|
| | ADI-R Social | ADI-R Verbal | ADI-R RRB | ADOS Total | ADOS Comm | ADOS Social | ADOS RRB | ADHD Index | Inattentive | Hyper/ Impulsive |
| Raw FC | 0.5610 | 0.5611 | 0.5611 | 0.2597 | 0.2597 | 0.2597 | 0.5228 | 0.5042 | 0.5162 | 0.5167 |
| Ours | **0.6049** | **0.6075** | **0.6051** | **0.6304** | **0.6302** | **0.6304** | **0.7501** | **0.5506** | **0.5663** | **0.5587** |

*Table 10.* Comparison of similarity definition locations on ABIDE and ADHD-200 (%).

| Method | ABIDE | | | | | ADHD-200 | | | | |
|---|---|---|---|---|---|---|---|---|---|---|
| | ACC | SEN | SPE | F1 | AUC | ACC | SEN | SPE | F1 | AUC |
| $\mathbf{E}_{\text{SI}}$ | 54.09±5.77 | 98.75±3.75 | 0.06±15.90 | 69.16±2.43 | 59.67±9.26 | 65.52±1.44 | 0±0 | 1±0 | 0±0 | 56.11±12.65 |
| Ours | 75.27±2.30 | 82.25±5.47 | 72.76±4.93 | 77.27±2.65 | 79.55±4.99 | 74.26±2.68 | 73.93±3.07 | 95.55±4.84 | 73.36±2.65 | 78.86±1.46 |

*Table 11.* Effect of stride on ABIDE and ADHD-200.

| Stride | ABIDE | | | | | ADHD-200 | | | | |
|---|---|---|---|---|---|---|---|---|---|---|
| | ACC | SEN | SPE | F1 | AUC | ACC | SEN | SPE | F1 | AUC |
| 10 | 68.32±2.47 | 75.18±5.63 | 66.45±4.58 | 70.28±3.12 | 72.15±3.54 | 67.45±2.91 | 66.32±6.18 | 79.18±5.52 | 68.12±3.41 | 68.23±4.28 |
| 20 | 69.87±2.42 | 75.94±5.58 | 68.12±4.49 | 71.56±3.05 | 72.38±3.42 | 68.92±2.85 | 66.85±6.05 | 80.96±5.18 | 70.48±3.32 | 69.56±4.05 |
| 30 | 71.23±2.38 | 78.45±5.49 | 70.26±4.41 | 72.29±2.95 | 75.02±3.31 | 70.26±2.78 | 69.13±5.92 | 83.15±5.03 | 70.55±3.25 | 70.82±3.87 |
| 40 | 72.18±2.35 | 80.61±5.38 | 70.48±4.35 | 73.58±2.84 | 76.45±3.18 | 71.05±2.72 | 70.48±5.81 | 83.28±4.82 | 71.38±3.18 | 72.93±3.65 |
| 50 | 72.86±2.32 | 80.85±5.31 | 71.05±4.28 | 74.52±2.75 | 77.68±3.05 | 71.92±2.68 | 71.52±5.69 | 84.12±4.85 | 72.15±3.02 | 74.57±3.42 |
| **Ours** | **75.27±2.30** | **82.25±5.47** | **72.76±4.93** | **77.27±2.65** | **79.55±4.99** | **74.26±2.68** | **73.93±3.07** | **95.55±4.84** | **73.36±2.65** | **78.86±1.46** |

Repetitive Behaviors (ADI-R RRB), as well as the Autism Diagnostic Observation Schedule (ADOS) including Total score, Communication (ADOS Comm), Social Interaction (ADOS Social), and Stereotyped Behaviors (ADOS RRB). For ADHD-200, we evaluate ADHD Index, Inattentive score, and Hyper/Impulsive score. We adopt connectome-based predictive modeling (CPM) method (Shen et al., 2017), where FC features are vectorized and selected ($p < 0.001$) via correlation analysis on the training set, followed by training a regression model under a 10-fold cross-validation protocol. Performance is evaluated using the correlation between predicted and observed scores, with permutation testing to assess statistical significance. The results in Tab. 9 show that our subject-invariant representations consistently outperform raw FC in prediction performance across all scales on both datasets. This demonstrates that the learned representations not only suppress subject variations but also preserve clinically meaningful and behaviorally relevant information, supporting their robustness.

### A.6. Comparison of Different Similarity Definition Locations

We compare defining similarity on $\mathbf{E}_{\text{SI}}$ and $\mathbf{Z}$. Results in Tab. 10 show that applying similarity on $\mathbf{E}_{\text{SI}}$ degrades performance by directly compressing the representation space, over-suppressing variation and causing collapse that removes disease-discriminative patterns. Applying inverse contrastive loss on $\mathbf{Z}$ is equivalent to imposing a structure-aware regularization on $\mathbf{E}_{\text{SI}}$, rather than directly compressing its representation space. This design enforces subject invariance while preserving the topological structure of FC, allowing $\mathbf{E}_{\text{SI}}$ to retain disease-relevant discriminative patterns.

### A.7. Validation of Temporal Splitting Strategy

Prior studies show that 30-60s windows suffice for stable FC estimation (Preti et al., 2017). Given our sequence lengths (190-502s for ABIDE and 205-600s for ADHD-200), each segment is sufficient for stable estimation. Moreover, our strategy is a special case of sliding-window modeling with both window length and stride equal to half the sequence. Varying the stride from 10 s to 50 s (Tab. 11) shows that smaller strides impair performance, as highly overlapping windows reduce effective contrast by producing near-duplicate samples, leading to vanishing gradients and hindering optimization. In contrast, non-overlapping segments are more independent, enabling effective learning of subject-invariant representations. We further compare with other augmentation methods, which mainly rely on perturbations such as node dropping, edge perturbation, and feature masking (Zhou et al., 2025). Results in Tab. 12 show that temporal splitting consistently achieves the best performance, as it preserves topology and biological semantics while producing natural same-subject pairs, which these perturbation methods fail to maintain.

*Table 12.* Performance (%) comparison of different augmentation strategies on ABIDE and ADHD-200.

| Method | ABIDE | | | | | ADHD-200 | | | | |
|---|---|---|---|---|---|---|---|---|---|---|
| | ACC | SEN | SPE | F1 | AUC | ACC | SEN | SPE | F1 | AUC |
| Node Dropping | 69.85±3.12 | 74.62±6.85 | 66.17±7.43 | 71.03±3.54 | 72.48±4.26 | 69.18±3.08 | 68.45±6.52 | 89.23±5.87 | 68.92±3.36 | 69.35±4.52 |
| Edge Perturbation | 71.41±2.96 | 77.88±6.21 | 69.02±6.95 | 74.92±3.21 | 74.15±3.88 | 70.56±2.89 | 69.82±6.18 | 89.67±5.54 | 69.15±3.21 | 70.72±4.18 |
| Feature Masking | 71.08±2.73 | 76.54±5.98 | 68.10±6.42 | 72.63±2.97 | 74.02±3.65 | 70.22±2.95 | 69.31±6.35 | 91.28±5.72 | 69.63±3.28 | 70.28±4.35 |
| **Ours** | **75.27±2.30** | **82.25±5.47** | **72.76±4.93** | **77.27±2.65** | **79.55±4.99** | **74.26±2.68** | **73.93±3.07** | **95.55±4.84** | **73.36±2.65** | **78.86±1.46** |

## A.8. Impact of Hierarchical Depth and Edge Retention

Our experiments indicate that the model achieves optimal performance under specific hierarchical configurations as shown in Fig. 7 and Fig. 8. In particular, the edge retention ratio $p = \{0.05, 0.15\}$ consistently yields the best results on both ABIDE and ADHD-200 datasets. Notably, when the hierarchical depth is less than two, performance drops significantly, indicating that a single-level topology is insufficient to capture subject-invariant representations. In contrast, increasing the depth beyond two does not lead to further gains, suggesting that excessive hierarchy introduces redundant structures rather than meaningful information.

The complete results of the two-level configuration are further presented in Fig. 9. For ABIDE, when $p_1$ is within the range from 0.01 to 0.09 and $p_2$ within 0.11 to 0.17, the performance remains stable regardless of the order of $p_1$ and $p_2$, while configurations outside this range show a noticeable decline. For ADHD-200, stable performance is observed over a broader range, with $p_1$ and $p_2$ from 0.11 to 0.25, as well as $p_1$ within 0.03 to 0.07 and $p_2$ within 0.13 to 0.17, again independent of the order. These results demonstrate that HTET maintains robust and stable performance across a wide spectrum of edge retention ratios. These analysis highlights the effectiveness of our hierarchical topology enhancement strategy, which enables HTET to extract subject-invariant patterns across multiple topological levels.

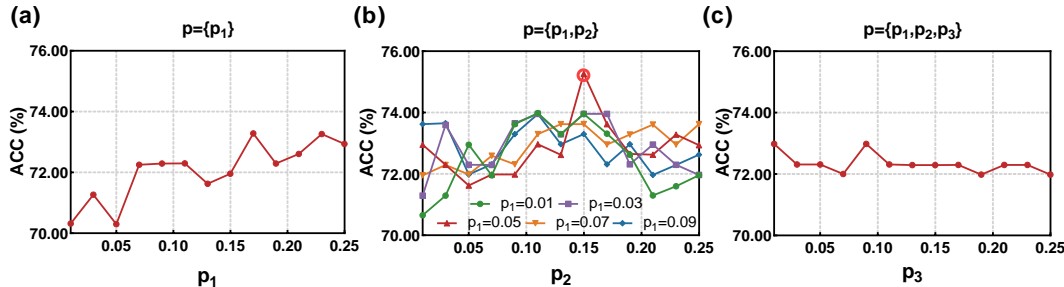

*Figure 7.* Identification accuracy (%) under varying hierarchical depth and edge retention ratios on ABIDE. (a) Single-level experiment with varying edge retention ratio $p_1$. (b) Two-level experiment with varying edge retention ratios $p_1$ and $p_2$. (c) Three-level experiment with varying $p_3$; for each $p_3$, accuracy is maximized over all $(p_1, p_2)$ combinations.

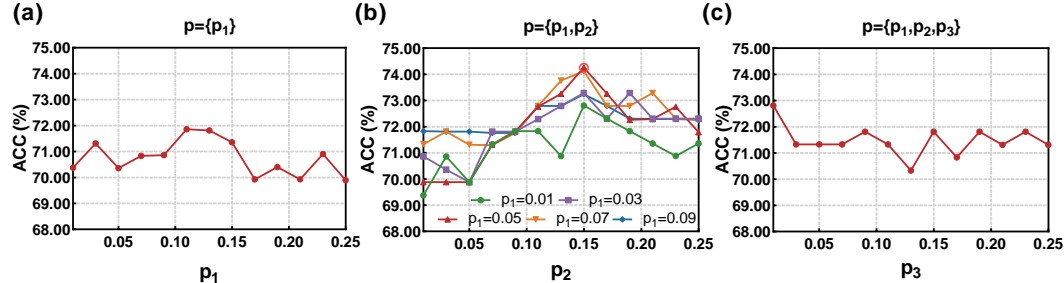

*Figure 8.* Identification accuracy (%) under varying hierarchical depth and edge retention ratios on ADHD-200. (a) Single-level experiment with varying edge retention ratio $p_1$. (b) Two-level experiment with varying edge retention ratios $p_1$ and $p_2$. (c) Three-level experiment with varying $p_3$; for each $p_3$, accuracy is maximized over all $(p_1, p_2)$ combinations.

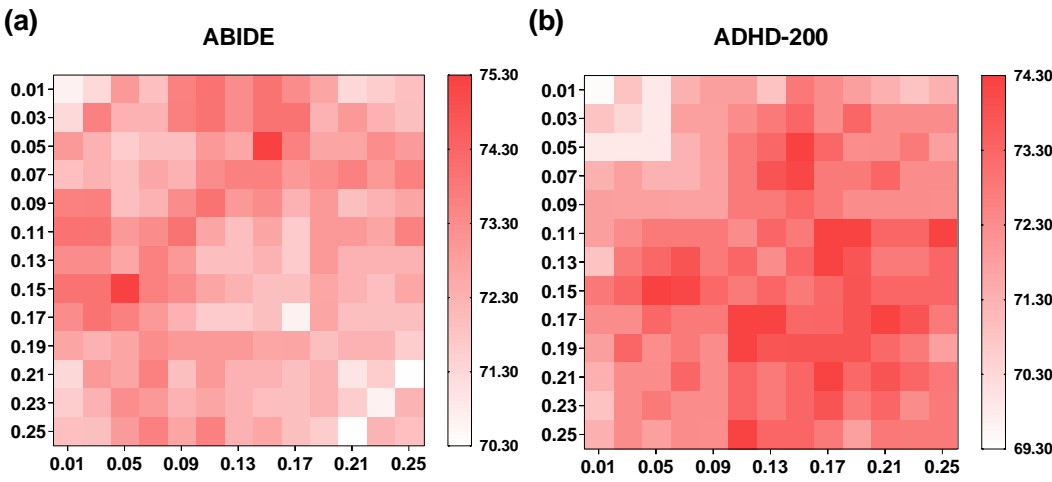

*Figure 9.* Complete results (ACC(%)) of two-level configuration on ABIDE and ADHD-200.

### A.9. Runtime and Memory Evaluation of HTET

We performed runtime and resource profiling to assess the computational overhead of HTET as shown in Tab. 13. Despite employing three parallel GT modules, corresponding to the original input plus the two-level configuration, the actual overhead remains moderate. Specifically, training time increases by approximately 1.25 times, inference time by 1.31 times, and memory usage by 2.46 times. These increases are substantially lower than the theoretical threefold overhead. More importantly, HTET delivers clear performance gains while incurring only modest overhead, with ACC improving by 3.97% on ABIDE. These results underscore the necessity and effectiveness of the HTET module in enhancing model performance.

*Table 13.* Comparison of runtime overhead between HTET and single-level configuration on ABIDE.

| Model | Training (s/epoch) | Inference (ms/sample) | Memory (MB) | ACC (%) |
|---|---|---|---|---|
| Single-level Configuration | 0.6892 | 0.6280 | 1697.66 | 71.30±5.16 |
| HTET | 0.8600 | 0.8240 | 4168.96 | 75.27±2.30 |

### A.10. Hyperparameter Analysis of Subject Invariance Loss

We conducted a comprehensive hyperparameter analysis of the proposed subject invariance loss on ABIDE (Fig. 10) and ADHD-200 (Fig. 11), examining the effects of the balancing coefficient $\beta$, the corrective term $\alpha$, and the temperature parameter $\tau$. For the balancing coefficient $\beta$, optimal performance is achieved at 0.85 for both ABIDE and ADHD-200. Specifically, ABIDE maintains stable performance when $\beta$ ranges from 0.8 to 0.95, while ADHD-200 remains stable when $\beta$ is between 0.8 and 0.9. Beyond these ranges, performance significantly declines, demonstrating the model's sensitivity to the trade-off between reconstructing the original graph and enforcing subject-invariance. For the corrective term $\alpha$, the optimal value is 1 for both datasets. ABIDE remains stable when $\alpha$ is between 1.0 and 1.2, and ADHD-200 remains stable when $\alpha$ is between 0.8 and 1.4, with significant drops observed outside these ranges. For the temperature parameter $\tau$, the model achieves the best performance at 0.1 for both ABIDE and ADHD-200, with notable performance degradation observed for values smaller or larger than 0.1. This highlights the importance of $\tau$ in controlling the sharpness of the similarity scaling and maintaining stable subject-invariant feature learning.

To examine the practical stability of the selected hyperparameters, we evaluate optimal hyperparameters across splits under 10-fold CV and and LOSO settings on ABIDE and ADHD-200, with selection performed on each split's training set. The results show high consistency, with small standard deviations (0.01) around $\beta \approx 0.85$, $\alpha \approx 1.00$, and $\tau \approx 0.10$. This indicates that the optimal region is stable across different data splits and evaluation protocols.

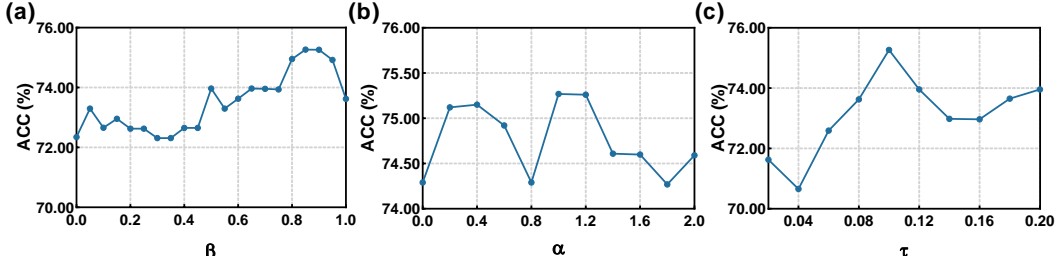

*Figure 10.* Hyperparameter analysis of subject invariance loss on ABIDE.

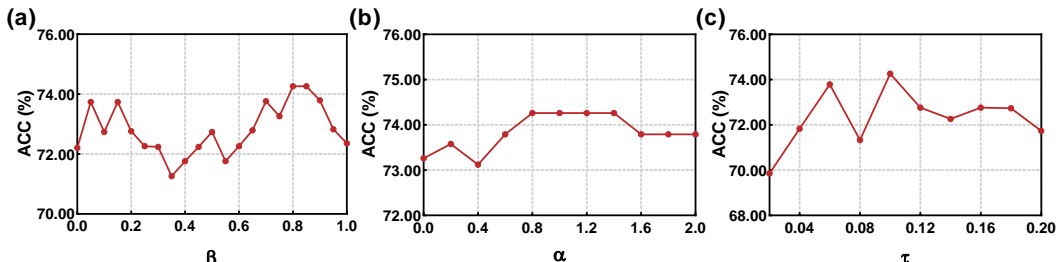

*Figure 11.* Hyperparameter analysis of subject invariance loss on ADHD-200.

## A.11. Impact of MHSA Depth and Number of Attention Heads

We investigated the impact of MHSA depth and the number of attention heads on both datasets (Fig. 12). Regarding MHSA depth, we found that shallow models consistently outperform deeper architectures across both datasets, indicating that for the given node sizes (ABIDE: 200, ADHD-200: 190), shallow models are more suitable. Deeper architectures may lead to overfitting in this setting. They could offer advantages in scenarios with higher-resolution brain networks, where the representational capacity of a shallow model may become insufficient. Concerning the number of attention heads, the choices are constrained by factors of the respective node counts (ABIDE: 200, ADHD-200: 190). Therefore, we evaluated 1, 2, 4, and 5 heads for ABIDE, and 1, 2, and 5 heads for ADHD-200. Optimal performance was achieved with 4 heads on ABIDE and 2 heads on ADHD-200, suggesting that excessive heads may introduce redundancy and computational overhead without further gains.

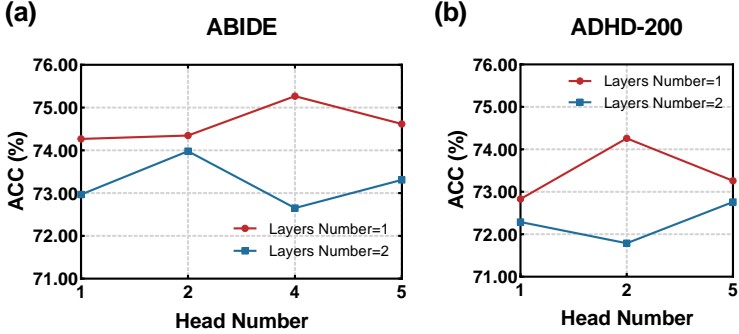

*Figure 12.* Impact of MHSA depth and number of attention heads.

## A.12. Impact of Reconstruction Module Configuration

We investigated the impact of the reconstruction module's depth and hidden dimension on classification performance. As shown in Tab. 14, increasing the depth from a single layer to two layers leads to a noticeable drop in ACC on both datasets, indicating that deeper architectures may introduce overfitting for the current node sizes. Regarding the hidden dimension, we studied a single-layer reconstruction module with dimensions ranging from 65 to 125 (Fig. 13). The ACC varied

non-monotonically with dimension, achieving optimal performance at 100 for both ABIDE and ADHD-200. This finding suggests that a moderate hidden dimension strikes a balance between sufficient representational capacity and avoiding overfitting, providing robust performance across datasets.

*Table 14.* Impact of reconstruction module's depth on ABIDE and ADHD-200.

| Architecture | ABIDE (ACC (%)) | ADHD-200 (ACC (%)) |
| --- | --- | --- |
| $200 \rightarrow 100$ | $75.27 \pm 2.30$ | – |
| $200 \rightarrow 100 \rightarrow 64$ | $70.65 \pm 5.73$ | – |
| $190 \rightarrow 100$ | – | $74.26 \pm 2.68$ |
| $190 \rightarrow 100 \rightarrow 64$ | – | $71.81 \pm 6.48$ |

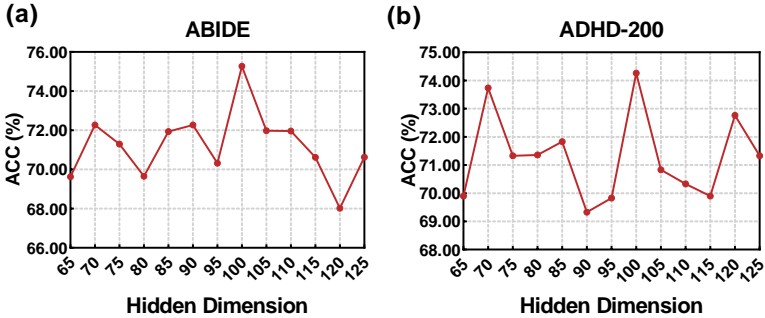

*Figure 13.* Effect of the reconstruction module's hidden dimension in a single layer.

### A.13. The Use of Large Language Models (LLMs)

In this work, we utilized Large Language Model to assist in polishing the manuscript. All scientific content and interpretations were independently authored by the research team.

