# OpenReview forum: "SI-IGCL: Subject Invariance-aware Inverse Graph Contrastive Learning for Psychiatric Disorder Identification"
_ICML.cc/2026/Conference — ICML 2026 regular_

### Official Review · Reviewer_1vA6 · 2026-03-11

**Soundness:** 3
**Presentation:** 2
**Significance:** 2
**Originality:** 3
**Overall Recommendation:** 4
**Confidence:** 3

**Summary:**

This paper proposes SI-IGCL for psychiatric disorder identification from functional brain networks. The paper inverses contrastive pre-training plus supervised fine-tuning, together with a hierarchical topology-enhanced transformer and a reconstruction constraint. Results on ABIDE and ADHD-200 show improved classification and cross-site generalization over previous methods.

**Compliance With Llm Reviewing Policy:**

Affirmed.

**Final Justification:**

After read the rebuttal and comments from other reviewers, I would like to maintain my current score.

**Key Questions For Authors:**

If possible, more discussion on generalization beyond ABIDE and ADHD-200 would strengthen the paper.

**Limitations:**

Yes

**Strengths And Weaknesses:**

Strength

- The motivation of this paper is clear. Handling inter-subject variability is a real challenge in neuroimaging-based diagnosis.

- The two-stage design, inverse contrastive objective, and reconstruction regularization are reasonable designed.

- The paper reports gains on both standard and cross-site settings, and also includes ablations and some interpretability analysis.

Weakness

- The theoretical and conceptual justification for attracting inter-subject embeddings while repelling intra-subject embeddings still feels somewhat counterintuitive. The paper argues that this helps remove subject-specific information, but it is not fully clear why this would not also weaken disease-relevant subject-level signals.

- The evaluation is still limited to two datasets in the same application domain. The paper would be better with broader validation or at least a clearer discussion of how well the learned invariance is expected to transfer across cohorts and preprocessing settings.

- The paper still not yet provide strong evidence that the learned subject-invariant features are especially meaningful or clinically robust.

---

> ### Author Rebuttal · Authors · 2026-03-30
>
> 1. Theoretical and empirical validation of disease-relevant information preservation
>
> Thank you for the insightful comment. We provide both theoretical justification and empirical analysis to verify whether the proposed objective preserves disease-relevant information. The results show that our subject-invariance objective enhances disease-relevant information relative to raw FC. Detailed discussion is provided in our response to Reviewer kF5e (Response 1).
>
> 2. Generalization of our model
>
> Thank you for the valuable comment. To evaluate generalization beyond ABIDE and ADHD-200, we conduct multi-class diagnostic experiments on ADHD-200 and ADNI. Specifically, we perform 4-class classification on ADHD-200, including typically developing control, ADHD Combined type, ADHD Inattentive type, and ADHD Hyperactive/Impulsive type. We further conduct 3-class classification on the Alzheimer’s Disease Neuroimaging Initiative (ADNI; https://adni.loni.usc.edu/) dataset, which comprises 442 subjects, including 175 normal controls, 153 mild cognitive impairment patients, and 114 Alzheimer’s disease patients. All methods are evaluated using a 10-fold cross-validation protocol, and performance is measured in terms of classification accuracy and Macro-F1 score. The results in Table s1 show that SI-IGCL achieves the best performance on most metrics across both datasets. These results demonstrate that the proposed model generalizes well across datasets and to more complex multi-class scenarios.
>
> Table s1: Multi-class classification performance (%) on ADHD-200 and ADNI.
> |Method|ADHD-200||ADNI||
> |-|-|-|-|-
> ||ACC|Macro-F1|ACC|Macro-F1
> |BrainGB|65.17±2.83|62.24±3.91|60.35±1.97|58.78±2.34
> |CRGNN|46.58±3.01|48.33±3.86|44.67±1.95|46.11±2.48
> |CI-GNN|64.32±2.76|57.85±3.71|60.71±1.92|60.44±2.26
> |CIA-GCL|58.73±2.88|52.51±3.78|60.28±2.01|58.52±2.44
> |BrainTrans|58.04±2.94|52.27±3.84|60.58±1.96|59.46±2.38
> |Com-BrainTF|64.95±2.80|59.73±3.74|59.36±1.89|56.51±2.31
> |METAFormer|63.62±2.72|50.08±3.66|59.63±1.93|58.77±2.29
> |Contrasformer|65.43±2.79|53.29±3.70|58.34±1.91|55.41±2.37
> |RGTNet|54.60±2.97|54.41±3.89|58.79±2.00|57.89±2.42
> |BrainIB|63.41±2.77|51.62±3.68|59.22±1.88|57.17±2.33
> |GBT|60.62±2.84|54.47±3.75|59.34±2.04|58.58±2.39
> |ALTER|56.78±2.91|54.89±3.82|60.07±1.95|59.42±2.41
> |LCCAF|51.96±2.86|53.80±3.76|60.42±1.92|57.92±2.36
> |CAGT|64.98±2.74|**64.71±3.67**|60.31±1.94|58.63±2.32
> |**SI-IGCL**|**67.43±2.71**|64.32±3.85|**63.58±2.03**|**62.15±2.46**
>
> 3. Validation of the meaningfulness and clinical robustness of learned representations
>
> Thank you for the thoughtful comment. To further validate the meaningfulness and clinical robustness of the learned subject-invariant representations, we conduct scale prediction experiments on ABIDE and ADHD-200. For ABIDE, we evaluate on the Autism Diagnostic Interview-Revised (ADI-R) including Social Interaction (ADI-R Social), Verbal Communication (ADI-R Verbal), and Restricted Repetitive Behaviors (ADI-R RRB), as well as the Autism Diagnostic Observation Schedule (ADOS) including Total score, Communication (ADOS Comm), Social Interaction (ADOS Social), and Stereotyped Behaviors (ADOS RRB). For ADHD-200, we evaluate ADHD Index, Inattentive score, and Hyper/Impulsive score. We adopt connectome-based predictive modeling (CPM) method [1], where FC features are vectorized and selected (p<0.001) via correlation analysis on the training set, followed by training a regression model under a 10-fold cross-validation protocol. Performance is evaluated using the correlation between predicted and observed scores, with permutation testing to assess statistical significance. The results in Table s2 show that our subject-invariant representations consistently outperform raw FC in prediction performance across all scales on both datasets. This demonstrates that the learned representations not only suppress subject variations but also preserve clinically meaningful and behaviorally relevant information, supporting their robustness.
>
> Table s2: Scale prediction performance (correlation coefficient) on ABIDE and ADHD-200.
> |Module||||ABIDE|||||ADHD-200||
> |-|-|-|-|-|-|-|-|-|-|-
> ||ADI-R Social|ADI-R Verbal|ADI-R RRB|ADOS Total|ADOS Comm|ADOS Social|ADOS RRB|ADHD Index|Inattentive|Hyper/Impulsive
> |Raw FC|0.5610|0.5611|0.5611|0.2597|0.2597|0.2597|0.5228|0.5042|0.5162|0.5167
> |Ours|0.6049|0.6075|0.6051|0.6304|0.6302|0.6304|0.7501|0.5506|0.5663|0.5587
>
> [1] Shen et al. Using connectome-based predictive modeling to predict individual behavior from brain connectivity[J]. nature protocols, 2017.

---

> > ### Author Rebuttal · Reviewer_1vA6 · 2026-04-06
> >
> > Thanks for the rebuttal, it addressed part of my concerns, I would like to maintain my current score.

---

> > > ### Author Response · Authors · 2026-04-07
> > >
> > > Thank you for the helpful feedback. We appreciate that our previous response addressed part of your concerns. In this revision, we provide further clarification and additional analysis to more directly address potential remaining concerns.
> > >
> > > **A1: (for Weakness 1)** Given that the orthogonality assumption between subject-specific and disease-related factors may not strictly hold in practice, we provide a derivation based on similarity evolution and gradient competition in the loss to show that disease-relevant patterns are preserved without relying on this assumption. We consider three types of pairs: same-subject pairs $P_{\text{same}}$ with similarity $S_{\text{same}}$, cross-subject intra-class pairs $P_{\text{within}}$ with similarity $S_{\text{within}}$, and cross-subject inter-class pairs $P_{\text{between}}$ with similarity $S_{\text{between}}$. The SI-IGCL objective consists of two components: an inverse contrastive loss $L_{IC}$ that repels same-subject pairs while attracting all cross-subject pairs, and a reconstruction loss $L_{R}$ that recovers the input graph to prevent representation collapse. Prior studies and empirical observations consistently show that the average similarity of same-subject pairs exceeds that of cross-subject pairs [1], with $\mathbb{E}[S_{\text{same}}^{(0)}] > \mathbb{E}[S_{\text{cross}}^{(0)}]$, and that the average similarity of cross-subject intra-class pairs is higher than that of cross-subject inter-class pairs [2-4], with $\mathbb{E}[S_{\text{within}}^{(0)}] > \mathbb{E}[S_{\text{between}}^{(0)}]$, in the raw FC space.
> > >
> > > For each anchor $z_i$, the gradient of $L_{IC}$ with respect to $z_i$ consists of two components. The repulsive term from same-subject pairs pushes $z_i$ away from its paired view and typically dominates the update due to their high initial similarity, thereby reducing reliance on subject-specific features. The attractive term pulls $z_i$ toward representations $z_j$ from other subjects, with weights determined by the current similarity. After suppressing subject-specific components, cross-subject similarity is primarily governed by disease-related features. Let $\mu_w(t)$ and $\mu_b(t)$ denote the mean similarity of cross-subject intra-class and inter-class pairs at time $t$, with $\mu_w(0) > \mu_b(0)$. This reflects that intra-class pairs have higher initial mean similarity and therefore receive larger attraction weights. The gradient dynamics are given by
> > > $$
> > > \frac{d\mu_w}{dt} \approx \eta (1 - \mu_w) w_w \quad \text{and} \quad \frac{d\mu_b}{dt} \approx \eta (1 - \mu_b) w_b,
> > > $$
> > > where $\eta$ denotes the learning rate, and $w_w$ and $w_b$ represent the effective gradient weights of cross-subject intra-class and inter-class pairs in the inverse contrastive loss. The weight ratio satisfies
> > > $$
> > > \frac{w_w}{w_b} = e^{(\mu_w - \mu_b)/\tau}.
> > > $$
> > > Therefore, the weight ratio grows exponentially with the similarity gap $\mu_w - \mu_b$, implying that the growth dynamics are dominated by the weighting mechanism. Since $\mu_w(0) > \mu_b(0)$, we have $w_w > w_b$ at initialization. As training proceeds, the increasing gap $\mu_w - \mu_b$ further amplifies the weight ratio, causing $\mu_w$ to grow consistently faster than $\mu_b$. The reconstruction loss enforces the recovery of the original graph structure and serves as a regularizer that promotes intra-class compactness while preserving sufficient within-class variability, thereby preventing representational collapse without undermining inter-class separability. The theoretical analysis above indicates that our objective further improves the separability of disease patterns.
> > >
> > > [1] Finn et al. Functional connectome fingerprinting: identifying individuals using patterns of brain connectivity[J]. Nature neuroscience, 2015.
> > >
> > > [2] Fu et al. Functional connectivity uniqueness and variability? Linkages with cognitive and psychiatric problems in children[J]. Nature Mental Health, 2023.
> > >
> > > [3] Zhang et al. Strength and similarity guided group-level brain functional network construction for MCI diagnosis[J]. Pattern recognition, 2019.
> > >
> > > [4] Tu et al. Identification of common neural substrates with connectomic abnormalities in four major psychiatric disorders: a connectome-wide association study[J]. European Psychiatry, 2021.
> > >
> > > **A2: (for Weakness 2 and Q1)** To evaluate generalization beyond ABIDE and ADHD-200, we conduct 3-class classification on ADNI and 4-class classification on ADHD-200. The results show that SI-IGCL achieves the best performance on most metrics, demonstrating strong generalization across cohorts and to more complex multi-class scenarios.
> > >
> > > **A3: (for Weakness 3)** To further validate the meaningfulness and clinical robustness of the learned subject-invariant representations, we conduct scale prediction experiments. The results show that our representations consistently outperform raw FC across all scales, indicating preservation of clinically and behaviorally relevant information.

---

### Official Review · Reviewer_4ENj · 2026-03-11

**Soundness:** 3
**Presentation:** 2
**Significance:** 3
**Originality:** 4
**Overall Recommendation:** 4
**Confidence:** 3

**Summary:**

This paper proposes SI-IGCL, a two-stage framework for psychiatric disorder identification from functional connectivity graphs. The key motivation is that inter-subject variability often dominates disease-related signals, which can cause models to learn subject-specific patterns instead of disorder-related representations. To address this issue, the authors design an inverse contrastive objective that repels representations from the same subject while attracting those from different subjects, encouraging subject-invariant features. A reconstruction loss is further introduced to prevent representation collapse. The model also incorporates a hierarchical topology enhanced transformer (HTET) to capture multi-level functional connectivity patterns. Experiments on ABIDE and ADHD-200 demonstrate improvements over several GNN- and graph-transformer-based baselines, along with ablation and interpretability analyses.

**Compliance With Llm Reviewing Policy:**

Affirmed.

**Final Justification:**

Thank you for the detailed rebuttal. The responses are helpful and address several of my questions. And I will keep my original recommendation of weak accept.

**Key Questions For Authors:**

1.Could the authors provide a stronger rationale or empirical justification for the temporal splitting strategy used to construct same-subject graph pairs? In particular, how does this strategy compare with other possible augmentation methods, and is there evidence that the non-overlapping split does not distort clinically relevant dynamic connectivity patterns?

2.Can the authors clarify the terminology and intuition behind the pair construction and inverse contrastive objective? The paper describes graphs from the same subject as “positive pairs” during construction, yet the learning objective later repels representations from the same subject.

3.How are the hyperparameters in the subject-invariance loss, like \alpha, \beta and \tao, selected in practice? Given the apparent sensitivity shown in the analysis, is there a principled strategy for choosing them without extensive grid search, especially under the 10-fold CV and LOSO settings?

4.Why is the similarity in Eq. (10) defined on the reconstructed matrices Z rather than directly on the latent representations E_{SI}? Did the authors compare these alternatives empirically, and if so, what motivated the final design choice?

5.Can the authors provide stronger evidence that the subject-invariance objective preserves disease-relevant information while removing subject-specific variation? Beyond the current NND/AFV/CKA analyses, additional evidence on label-relevant signal retention would further strengthen the central claim.

**Limitations:**

Yes

**Strengths And Weaknesses:**

Strengths

1. The paper addresses an important challenge in neuroimaging-based disorder identification: inter-subject variability often dominates disease-related signals, which can cause models to learn subject-specific patterns instead of clinically meaningful representations. The proposed subject-invariant learning objective is therefore well aligned with the problem.

2. The inverse contrastive objective that repels representations from the same subject while attracting those from different subjects is an interesting idea for encouraging subject-invariant features. The authors also recognize the potential risk of representation collapse and introduce a graph reconstruction loss as an architectural constraint, which is validated through ablation studies.

3. The paper evaluates the method on two benchmark datasets (ABIDE and ADHD-200) and includes multiple baselines, ablation studies, and cross-site experiments. In particular, the leave-one-site-out  evaluation demonstrates improved cross-site generalization under heterogeneous data distributions.

Weaknesses

1. The positive pairs used for contrastive learning are constructed by splitting the fMRI time series into two non-overlapping segments. Given that functional connectivity exhibits complex temporal dynamics, the paper does not sufficiently analyze whether this simple temporal partition may alter or disrupt important dynamic connectivity patterns.

2. The subject invariance loss introduces additional hyperparameters (e.g., the corrective term \alpha and balance coefficient \beta). The result show that performance can drop significantly when these parameters deviate from their optimal ranges, raising concerns about the robustness of the method.

3. The paper describes graphs from the same subject as positive pairs during construction, yet the learning objective later repels representations from the same subject. This terminology may cause confusion and could benefit from clearer explanation.

---

> ### Author Rebuttal · Authors · 2026-03-30
>
> 1. Validation of temporal splitting strategy
>
> Thank you for the insightful comment. Temporal splitting is widely adopted in fMRI contrastive learning to construct same-subject graph pairs [1]. The concern is orthogonal to our setting. Dynamic connectivity analysis partitions time series into windows to model temporal variations of FC, whereas our construction of same-subject pairs is independent of temporal dependencies and depends on whether the two FC graphs originate from the same subject and FC stability. Prior studies show that 30-60s windows suffice for stable FC estimation [2]. Given our sequence lengths (190-502s for ABIDE and 205-600s for ADHD-200), each segment is sufficient for stable estimation. Moreover, our strategy is a special case of sliding-window modeling with both window length and stride equal to half the sequence. Varying the stride from 10 s to 50 s (Table s1) shows that smaller strides impair performance, as highly overlapping windows reduce effective contrast by producing near-duplicate samples, leading to vanishing gradients and hindering optimization. In contrast, non-overlapping segments are more independent, enabling effective learning of subject-invariant representations. We further compare with other augmentation methods, which mainly rely on perturbations such as node dropping, edge perturbation, and feature masking [3]. Results in Table s2 show that temporal splitting consistently achieves the best performance, as it preserves topology and biological semantics while producing natural same-subject pairs, which these perturbation methods fail to maintain. Tables s1 and s2 are at https://anonymous.4open.science/r/SI-IGCL_rebuttal.
>
> [1] Zhou et al. GIN-Transformer based Pairwise Graph Contrastive Learning Framework[J]. Neural Networks, 2026.
>
> [2] Preti et al. The dynamic functional connectome: State-of-the-art and perspectives[J]. Neuroimage, 2017.
>
> [3] Zhou et al. Data augmentation on graphs: A technical survey[J]. ACM Computing Surveys, 2025.
>
> 2. Clarification of pair terminology and objective
>
> Thank you for the constructive comment. We appreciate your use of “same-subject pairs” in the first question, which improved the terminology. We will replace “positive pairs” with “same-subject pairs” and “negative pairs” with “cross-subject pairs” to avoid confusion. Our inverse contrastive objective repels same-subject pairs while attracting cross-subject pairs to encourage subject-invariant representations, and the reconstruction loss prevents representation collapse. Results in Table 4 of the paper show that the learned embeddings effectively reduce similarity within same-subject pairs and increase similarity across cross-subject pairs, yielding robust and subject-invariant representations.
>
> 3. Adaptive hyperparameter selection strategy
>
> Thank you for the helpful comment. To avoid exhaustive grid search, we design an adaptive hyperparameter strategy inspired by prior work [1], formulating the inverse contrastive loss and reconstruction loss as competing objectives within a multi-objective optimization framework. Periodic non-dominated sorting is applied to approximate the Pareto front, and $\beta$ is updated via variance-aware relative improvement (VARI). $\tau$ is adaptively scaled by the loss magnitude for stable similarity calibration, while $\alpha$ follows the same VARI-guided dynamics. This adaptive strategy can be seamlessly integrated into 10-fold CV and LOSO settings. Empirically, it converges to stable regions near grid-search optima across datasets. $\beta$ stabilizes around 0.82–0.92, $\alpha$ around 0.95–1.22, and $\tau$ around 0.09–0.11, avoiding exhaustive tuning while maintaining strong generalization.
>
> [1] Wu et al. A Multi-Objective Optimization Framework for Adaptive Weighting in Physics-Informed Machine Learning[C]. AAAI, 2026.
>
> 4. Similarity definition on $Z$.
>
> Thank you for the thoughtful comment. We compare defining similarity on $E_{SI}$ and $Z$. Results in Table s3 (https://anonymous.4open.science/r/SI-IGCL_rebuttal) show that applying similarity on $E_{SI}$ degrades performance by directly compressing the representation space, over-suppressing variation and causing collapse that removes disease-discriminative patterns. In contrast, we define similarity on $Z$, where $Z = Decoder(E_{SI})$. Applying inverse contrastive loss on $Z$ is equivalent to imposing a structure-aware regularization on $E_{SI}$, rather than directly compressing its representation space. This design enforces subject invariance while preserving the topological structure of FC, allowing $E_{SI}$ to retain disease-relevant discriminative patterns.
>
> 5. Validation of disease-relevant information retention
>
> Thank you for the insightful comment. We provide theoretical and empirical evidence that the proposed objective enhances disease-relevant information over raw FC. Detailed discussion is provided in our response to Reviewer kF5e (Response 1).

---

> > ### Author Rebuttal · Reviewer_4ENj · 2026-04-02
> >
> > Thank you for the detailed rebuttal. The responses are helpful and address several of my questions. However, for hyperparameter selection, the rebuttal proposes an adaptive strategy, but this does not appear to match the procedure described in the paper itself, where the hyperparameters are instead presented as fixed values chosen through systematic tuning and analyzed via sensitivity experiments. That said, the rebuttal is constructive and improves the presentation of the paper. My overall assessment remains unchanged, and I therefore keep my original recommendation of weak accept.

---

> > > ### Author Response · Authors · 2026-04-04
> > >
> > > Thank you for the helpful and constructive feedback. We sincerely apologize for any confusion. We clarify that the hyperparameters are chosen through systematic tuning and evaluated via sensitivity analysis in the current paper. To examine their practical stability, we evaluate optimal hyperparameters across splits under 10-fold CV and and LOSO settings on ABIDE and ADHD-200 (Table s1), with selection performed on each split’s training set. The results show high consistency, with small standard deviations around $\beta \approx 0.85$, $\alpha \approx 1.00$, and $\tau \approx 0.10$. This indicates that the optimal region is stable across different data splits and evaluation protocols. The small standard deviations suggests that the hyperparameters are well-constrained in practice, improving usability in real-world deployment. This suggests that sensitivity reflects a narrow but stable optimum rather than instability in parameter selection.
> > >
> > > Table s1: Optimal hyperparameters across splits under 10-fold CV and LOSO settings on ABIDE and ADHD-200 (mean±std).
> > > |Hyperparameter|ABIDE||ADHD-200||
> > > |-|-|-|-|-
> > > ||10-fold CV|LOSO settings|10-fold CV|LOSO settings
> > > |$\beta$|0.85±0.03|0.85±0.01|0.85±0.01|0.85±0.02
> > > |$\alpha$|1.00±0.01|1.00±0.01|1.00±0.02|1.00±0.01
> > > |$\tau$|0.10±0.00|0.10±0.01|0.10±0.01|0.10±0.00
> > >
> > > To explore a principled strategy for choosing hyperparameters without extensive grid search, we further investigated an adaptive hyperparameter selection approach as an extension beyond the current paper, motivated by prior work [1]. The core idea of this strategy is to leverage the principled role of each hyperparameter to guide adaptive updates. Specifically, $\beta$ balances the inverse contrastive loss and reconstruction loss, reflecting a trade-off between suppressing subject-specific variation and preserving topological discriminative information. We formulate training as a multi-objective optimization problem and perform periodic non-dominated sorting on the two losses. We update $\beta$ using variance-aware relative improvement (VARI) to drive it toward the Pareto front without manual tuning. The corrective term $\alpha$ is designed to alleviate early optimization stagnation. We adapt $\alpha$ based on the similarity distribution of cross-subject pairs. When the similarity mean exceeds a threshold, $\alpha$ increases to promote informative cross-subject pairs with lower similarity and maintain meaningful gradient signals for sustained loss reduction. When similarity is low, $\alpha$ decreases. The temperature $\tau$ controls the sharpness of the similarity distribution. We adopt a loss-scale feedback mechanism that monitors the magnitude of the inverse contrastive loss. When the loss is large, $\tau$ increases to soften the distribution. When the loss is small, $\tau$ decreases to sharpen it. VARI is used to smooth fluctuations and allows $\tau$ to stabilize in a regime with strong contrastive signals. Empirically, we observe that the adaptive strategy converges to stable regions. $\beta$ stabilizes around 0.82–0.92, $\alpha$ around 0.95–1.22, and $\tau$ around 0.09–0.11. These ranges are close to the grid search optima, and the resulting model achieves comparable performance to the fixed hyperparameter version. Our adaptive strategy is an extension inspired by the first-round feedback and aims to provide a principled alternative to exhaustive grid search.
> > >
> > > [1] Wu et al. A Multi-Objective Optimization Framework for Adaptive Weighting in Physics-Informed Machine Learning[C]. AAAI, 2026.

---

### Official Review · Reviewer_kF5e · 2026-03-12

**Soundness:** 4
**Presentation:** 4
**Significance:** 3
**Originality:** 3
**Overall Recommendation:** 4
**Confidence:** 3

**Summary:**

This paper proposes SI-IGCL (Subject Invariance-aware Inverse Graph Contrastive Learning) for psychiatric disorder identification from functional brain networks derived from fMRI. The key challenge addressed is the inter-subject variability, which often harms generalization in brain connectivity analysis. To solve this problem, the authors introduced a two-stage paradigm: pretraining to encourage the model to learn subject-invariant representation, and supervised fine-tuning for disease identification. Experiments show the effectiveness of SI-IGCL on psychiatric disorder identification.

**Compliance With Llm Reviewing Policy:**

Affirmed.

**Final Justification:**

Thank you for providing the theoretical analysis. My concerns have been addressed. I will keep my original score.

**Key Questions For Authors:**

• The SI-IGCL objective enforces subject invariance by repelling intra-subject embeddings. However, some subject-specific characteristics may correlate with disease patterns. It is therefore unclear whether the method removes only nuisance variability or also suppresses useful pathology-related information. Can the authors provide theoretical justification or empirical analysis for this?
• The current experiments appear to focus primarily on binary classification tasks. In practical clinical settings, psychiatric and neurological diagnosis often involves multi-class problems (e.g., multiple ADHD subtypes or different stages of Alzheimer’s disease in ADNI).
How well would SI-IGCL generalize to more complex multi-class diagnostic settings (e.g., 4-class ADHD or multi-stage ADNI classification)? Have the authors considered evaluating the method under such scenarios?

**Limitations:**

Yes

**Strengths And Weaknesses:**

Strengths:
• The authors tackle an important but undervalued problem that substantial inter-subject differences in brain data may contradict the normal group homogeneity assumption.
• The HTET module extends transformer-style attention to capture hierarchical brain network structures, equipping SI-IGCL with the ability to capture both local and global structure information.
• The experimental results are quite good.

Weaknesses:
• The training objective of SI-IGCL is very complex.
• The SI-IGCL has the risk of destroying useful subject-specific signals.

---

> ### Author Rebuttal · Authors · 2026-03-30
>
> 1. Validation of disease-relevant information retention
>
> Thank you for the insightful comment. Neuroimaging studies show that ASD and ADHD exhibit consistent disease-related connectivity patterns across subjects, with abnormal default mode network connectivity as a hallmark of both disorders [1]. Prior studies by Yamashita et al. and Diao et al. show that subject-level variations are largely driven by non-pathological factors and are approximately orthogonal to disease-relevant patterns [2][3]. The original functional connectivity graph $G$ contains both subject-specific information $I(G; S)$ and disease-related information $I(G; Y)$. The inverse contrastive objective guides the encoder $f: G \to E_{SI}$ to minimize subject-specific information $I(E_{SI}; S)$, thereby removing subject variation. The reconstruction loss $\lVert Decoder(E_{SI}) - G \rVert_F^2$ preserves input information by maximizing $I(E_{SI}; G)$. Since disease-related patterns are shared across subjects and are approximately orthogonal to subject-specific variation, minimizing $I(E_{SI}; S)$ under the constraint that $I(E_{SI}; G)$ is lower bounded is equivalent to maximizing disease-related information $I(E_{SI}; Y)$. Therefore, the proposed objective does not remove disease-relevant information.
>
> Empirically, we directly quantify disease-related signal retention using two metrics. Linear classifier retention (LCR) is defined as the ratio of accuracy gains over chance achieved by the same linear classifier on the latent representation versus raw FC, reflecting linear separability of disease-related signals. Mutual information retention (MIR) is defined as the ratio between the mutual information of the representation and labels and that of raw FC and labels, estimated using a k-nearest neighbor estimator [4]. If LCR and MIR are equal to or greater than 1, disease-relevant information is preserved or enhanced relative to raw FC, while values below 1 indicate degradation. As shown in Table s1, both LCR and MIR are consistently greater than 1 on ABIDE and ADHD-200, indicating that disease-relevant information is enhanced. In addition, t-SNE visualization (Fig. s1, available at https://anonymous.4open.science/r/SI-IGCL_rebuttal) shows that the learned representations form better-separated clusters aligned with disease labels, further supporting improved discriminability. Moreover, to validate the clinical relevance and robustness of the subject-invariant representations, we conduct scale prediction experiments. The results show that our representations consistently outperform raw FC across all scales on both datasets, indicating preservation of clinically and behaviorally relevant information. Details are provided in our response to Reviewer 1vA6 (Response 3).
>
> Table s1: Disease-related information retention on ABIDE and ADHD-200.
> |Dataset|LCR|MIR
> |-|-|-
> |ABIDE|1.428|1.529
> |ADHD-200|1.663|1.227
>
> [1] Harikumar et al. A review of the default mode network in autism spectrum disorders and attention deficit hyperactivity disorder[J]. Brain connectivity, 2021.
>
> [2] Yamashita et al. Computational mechanisms of neuroimaging biomarkers uncovered by multicenter resting-state fMRI connectivity variation profile[J]. Molecular Psychiatry, 2025.
>
> [3] Diao et al. Heterogeneity-Aware, Multiscale Annotation of Shared and Specific Neurobiological Signatures among Major Neurodevelopmental Disorders[J]. Research, 2026.
>
> [4] Pohl et al. Clarifying the conceptual dimensions of representation in neuroscience[J]. Nature Reviews Neuroscience, 2026.
>
> 2. Multi-class generalization evaluation
>
> Thank you for the valuable comment. To evaluate generalization to multi-class diagnostic settings, we conduct 4-class classification on ADHD-200 and 3-class classification on ADNI. Details are provided in our response to Reviewer 1vA6 (Response 2). The results show that SI-IGCL achieves the best performance on most metrics across both datasets, demonstrating strong generalization to more complex multi-class scenarios.
>
> 3. The training objective of SI-IGCL is very complex
>
> Thank you for highlighting this point. Substantial inter-subject variability pose a major challenge in psychiatric diagnosis, and single supervised or conventional contrastive objectives often capture spurious correlations driven by individual variability. This motivates a multi-objective strategy that preserves disease-discriminative information while suppressing subject-specific variation. Our inverse contrastive objective repels same-subject pairs while attracting cross-subject pairs to encourage subject-invariant representations, and the reconstruction loss prevents representation collapse. Results in Table 4 of the paper show that the learned embeddings effectively reduce similarity within same-subject pairs and increase similarity across cross-subject pairs, yielding robust and subject-invariant representations.

---

> > ### Author Rebuttal · Reviewer_kF5e · 2026-04-03
> >
> > The response provides useful empirical evidence (LCR/MIR) suggesting that disease-relevant information is retained, which partially addresses the concern. However, the theoretical justification relies on a strong orthogonality assumption between subject-specific and disease-related factors, which may not hold in practice, making the argument less convincing. Overall, the concern is mitigated empirically but not fully resolved from a theoretical standpoint.

---

> > > ### Author Response · Authors · 2026-04-04
> > >
> > > Thank you for the helpful feedback. We agree that the orthogonality assumption between subject-specific and disease-related factors may not strictly hold in practice.  To address this concern, we provide a derivation based on similarity evolution and gradient competition in the loss to show that disease-relevant patterns are preserved without relying on the orthogonality assumption. We consider three types of pairs: same-subject pairs $P_{\text{same}}$ with similarity $S_{\text{same}}$, cross-subject intra-class pairs $P_{\text{within}}$ with similarity $S_{\text{within}}$, and cross-subject inter-class pairs $P_{\text{between}}$ with similarity $S_{\text{between}}$. The SI-IGCL objective consists of two components: an inverse contrastive loss $L_{IC}$ that repels same-subject pairs while attracting all cross-subject pairs, and a reconstruction loss $L_{R}$ that recovers the input graph to prevent representation collapse. Prior studies and empirical observations consistently show that the average similarity of same-subject pairs exceeds that of cross-subject pairs [1], with $\mathbb{E}[S_{\text{same}}^{(0)}] > \mathbb{E}[S_{\text{cross}}^{(0)}]$, and that the average similarity of cross-subject intra-class pairs is higher than that of cross-subject inter-class pairs [2-4], with $\mathbb{E}[S_{\text{within}}^{(0)}] > \mathbb{E}[S_{\text{between}}^{(0)}]$, in the raw FC space.
> > >
> > > For each anchor $z_i$, the gradient of $L_{IC}$ with respect to $z_i$ consists of two components. The repulsive term from same-subject pairs pushes $z_i$ away from its paired view and typically dominates the update due to their high initial similarity, thereby reducing reliance on subject-specific features. The attractive term pulls $z_i$ toward representations $z_j$ from other subjects, with weights determined by the current similarity. After suppressing subject-specific components, cross-subject similarity is primarily governed by disease-related features. Let $\mu_w(t)$ and $\mu_b(t)$ denote the mean similarity of cross-subject intra-class and inter-class pairs at time $t$, with $\mu_w(0) > \mu_b(0)$. This reflects that intra-class pairs have higher initial mean similarity and therefore receive larger attraction weights. The gradient dynamics are given by
> > > $$
> > > \frac{d\mu_w}{dt} \approx \eta (1 - \mu_w) w_w \quad \text{and} \quad \frac{d\mu_b}{dt} \approx \eta (1 - \mu_b) w_b,
> > > $$
> > > where $\eta$ denotes the learning rate, and $w_w$ and $w_b$ represent the effective gradient weights of cross-subject intra-class and inter-class pairs in the inverse contrastive loss. The weight ratio satisfies
> > > $$
> > > \frac{w_w}{w_b} = e^{(\mu_w - \mu_b)/\tau}.
> > > $$
> > > Therefore, the weight ratio grows exponentially with the similarity gap $\mu_w - \mu_b$, implying that the growth dynamics are dominated by the weighting mechanism. Since $\mu_w(0) > \mu_b(0)$, we have $w_w > w_b$ at initialization. As training proceeds, the increasing gap $\mu_w - \mu_b$ further amplifies the weight ratio, causing $\mu_w$ to grow consistently faster than $\mu_b$. The reconstruction loss enforces the recovery of the original graph structure and serves as a regularizer that promotes intra-class compactness while preserving sufficient within-class variability, thereby preventing representational collapse without undermining inter-class separability. The theoretical analysis above indicates that our objective further improves the separability of disease patterns.
> > >
> > > [1] Finn et al. Functional connectome fingerprinting: identifying individuals using patterns of brain connectivity[J]. Nature neuroscience, 2015.
> > >
> > > [2] Fu et al. Functional connectivity uniqueness and variability? Linkages with cognitive and psychiatric problems in children[J]. Nature Mental Health, 2023.
> > >
> > > [3] Zhang et al. Strength and similarity guided group-level brain functional network construction for MCI diagnosis[J]. Pattern recognition, 2019.
> > >
> > > [4] Tu et al. Identification of common neural substrates with connectomic abnormalities in four major psychiatric disorders: a connectome-wide association study[J]. European Psychiatry, 2021.

---

### Decision · Program_Chairs · 2026-04-30

**Decision:**

Accept (regular)

**Comment:**

This paper proposes the Subject Invariance-aware Inverse Graph Contrastive Learning (SI-IGCL) model with a two-stage paradigm: self-supervised subject-invariant pre-training and supervised fine-tuning for identification. It introduces an inverse contrastive objective, a structure-preserving reconstruction constraint, and a Hierarchical Topology Enhanced Transformer. All three reviewers provide positive feedback by recognizing the clear motivations, significant technical contributions, and extensive evaluations. In the rebuttal, most concerns from reviewers are resolved.